# CUTseq is a versatile method for preparing multiplexed DNA sequencing libraries from low-input samples

Xiaolu Zhang[1], Silvano Garnerone[1], Michele Simonetti[1], Luuk Harbers [1], Marcin Nicoś [1,2], Reza Mirzazadeh[1], Tiziana Venesio[3], Anna Sapino [3,4], Johan Hartman [5,6], Caterina Marchiò[3,4], Magda Bienko [1,7]* & Nicola Crosetto [1,7]*

Current multiplexing strategies for massively parallel sequencing of genomic DNA mainly rely on library indexing in the final steps of library preparation. This procedure is costly and time-consuming, because a library must be generated separately for each sample. Furthermore, library preparation is challenging in the case of fixed samples, such as DNA extracted from formalin-fixed paraffin-embedded (FFPE) tissues. Here we describe CUTseq, a method that uses restriction enzymes and in vitro transcription to barcode and amplify genomic DNA prior to library construction. We thoroughly assess the sensitivity and reproducibility of CUTseq in both cell lines and FFPE samples, and demonstrate an application of CUTseq for multi-region DNA copy number profiling within single FFPE tumor sections, to assess intratumor genetic heterogeneity at high spatial resolution. In conclusion, CUTseq is a versatile and cost-effective method for library preparation for reduced representation genome sequencing, which can find numerous applications in research and diagnostics.

[1] Science for Life Laboratory, Department of Medical Biochemistry and Biophysics, Karolinska Institutet, Stockholm SE-17165, Sweden. [2] Department of Pneumonology, Oncology and Allergology, Medical University of Lublin, 20954 Lublin, Poland. [3] Pathology Unit, Candiolo Cancer Institute, FPO-IRCCS, 10060 Candiolo (TO), Italy. [4] Department of Medical Sciences, University of Turin, Turin, Italy. [5] Department of Oncology and Pathology, Karolinska Institutet, Stockholm SE-17177, Sweden. [6] Department of Clinical Pathology, Karolinska University Laboratory, 17176 Stockholm, Sweden. [7] These authors contributed equally: Magda Bienko, Nicola Crosetto. *email: magda.bienko@ki.se; nicola.crosetto@ki.se

In the past decade, next-generation sequencing (NGS) technologies have become widely available in diagnostics and research laboratories[1,2]. During this time, the number of methodologies for preparing DNA libraries for NGS has greatly expanded, whereas the cost of sequencing has exponentially dropped[1,2]. In spite of this progress, sequencing of multiple samples in parallel remains costly, mainly due to the way in which multiplexing is achieved. Typically, this is done by indexing libraries prepared from individual samples, followed by pooling together multiple libraries in the same sequencing run. This means that all the steps in the library preparation procedure must be repeated for every sample to be sequenced, which is labor-intensive and multiplies the cost of reagents. Furthermore, accurate normalization of library concentration is necessary before multiple libraries can be pooled together, which is not always possible and requires additional reagents. In contrast, being able to directly barcode genomic DNA (gDNA) prior to library construction, followed by pooling of differentially barcoded samples into a single library, should enable high levels of multiplexing at much lower cost.

An example of application that would greatly benefit from improved solutions for NGS library multiplexing is multi-region DNA sequencing of tumor samples[3]. In this approach, DNA is extracted from multiple regions within the same tumor mass, or from multiple tumor sites in the same patient, and a library is prepared for each region. Multi-region tumor sequencing has been successfully used to assess levels of intratumor heterogeneity and to infer tumor evolution in different cancer types[3]. One limitation of current multi-region tumor sequencing approaches is the size of the regions examined, which must be sufficiently large to enable the recovery of enough DNA to construct a library from every region separately. This precludes the possibility of examining a larger number of smaller regions, e.g., within a single tissue section, which would enable assessing intratumor heterogeneity at much higher spatial resolution. This, together with the high cost needed to make a single library for every region sampled, currently limits the applicability of multi-region tumor sequencing in routine cancer diagnostics.

Several approaches have been developed to barcode gDNA as well as to amplify sub-nanogram amounts of gDNA prior to library preparation. Direct incorporation of sequencing adapters into gDNA by engineered transposases allows rapid library preparation and is the basis of successful commercial solutions such as Nextera from Illumina, Inc. However, this approach still requires that individual libraries are generated from each sample, and then pooled together before sequencing. On the other hand, whole-genome amplification methods, such as DOP-PCR[4], MDA[5], MALBAC[6], and the more recent SCMDA[7] and LIANTI[8], achieve direct gDNA barcoding during genome amplification, so that multiple samples can be pooled together into a single multiplexed library. Although such methods are specifically tailored for whole-genome sequencing of single cells, they could, in principle, also be used for other applications, for instance to generate multiplexed libraries for multi-region tumor sequencing in tissue sections. One limitation, however, is that whole-genome amplification requires intact DNA and thus is problematic in fixed tissue samples, in particular formalin-fixed, paraffin-embedded (FFPE) specimens, which still represent a cornerstone in pathology. In addition, whole-genome amplification methods are very costly, making them hardly applicable to routine diagnostics.

To overcome these limitations, here we develop a method, which we name CUTseq, that combines restriction endonucleases with in vitro transcription (IVT), to construct highly multiplexed DNA libraries for reduced representation genome sequencing of multiple samples in parallel. We show that CUTseq can be used to barcode gDNA extracted from both non-fixed and fixed samples, including old archival FFPE tissue sections. We benchmark CUTseq by comparing it with a widely used method of DNA library preparation and demonstrate that CUTseq can be used for reduced representation genome and exome sequencing, enabling reproducible DNA copy number profiling and single-nucleotide variant (SNV) calling in both cell and low-input FFPE tissue samples. We then show an application of CUTseq for assessing intratumor genetic heterogeneity, by profiling DNA copy number levels in multiple small regions of individual FFPE tumor sections. Lastly, we describe a workflow for rapid and cost-effective preparation of highly multiplexed CUTseq libraries, which can be applied in the context of high-throughput genetic screens and for cell line authentication.

## Results

**CUTseq workflow**. We aimed at developing a versatile method for preparing highly multiplexed DNA sequencing libraries, by barcoding gDNA from multiple samples directly after purification. To this end, we devised the CUTseq workflow as depicted in Fig. 1a. The procedure starts by digesting gDNA extracted from either non-fixed or fixed samples, including low-input FFPE tissue specimens, using a type-II restriction endonuclease that leaves staggered ends. After gDNA is digested, the restricted sites are ligated to specialized double-stranded DNA adapters that contain a sample-specific barcode sequence, a unique molecular identifier (UMI)[9], the RA5 Illumina sequencing adapter, and the T7 promoter sequence. After ligation, multiple samples are pooled together and the genomic sequences flanking the ligated restriction sites are amplified using IVT by the T7 RNA polymerase. Lastly, a sequencing library is generated from the IVT product, based on the small RNA library preparation kit from Illumina (Methods). A step-by-step CUTseq protocol is available in the Supplementary Methods and at Protocol Exchange (https://doi.org/10.21203/rs.2.1742/v1). The sequences of all the CUTseq adapters used in this study are provided in Supplementary Data 1.

**CUTseq implementation**. We first tested the feasibility of CUTseq by constructing libraries from gDNA extracted from five different human cancer cell lines and IMR90 primary human fibroblasts (Methods). We digested the samples using either a more frequent four-base cutter (NlaIII) or a less frequent six-base cutter (HindIII). We selected the enzymes among a list of commercially available restriction enzymes that leave staggered DNA ends and are methylation insensitive (Supplementary Table 1), choosing the least expensive enzymes with the most homogeneous distribution of recognition sites in the human genome (Supplementary Fig. 1a-d). The distance between two consecutive recognition sites is $210 \pm 286$ bp and $3422 \pm 3684$ bp (mean ± SD) for NlaIII and HindIII, respectively. We sequenced all the libraries on the NextSeq 500 platform from Illumina, Inc., and processed the reads through a custom-made pipeline that we make freely available (Supplementary Software). All the libraries showed a homogeneous fragment size distribution and yielded a high proportion of reads with the expected prefix ($95\% \pm 0.01\%$, mean ± SD), high mappability ($96.58\% \pm 0.02\%$, mean ± SD), very low rate of sequencing errors ($0.81\% \pm 0.001\%$, mean ± SD), even partitioning between the Watson and Crick strands, and balanced distribution of all the four bases at every position along the UMI sequence (Supplementary Fig. 2a-c and Supplementary Data 2). In the IMR90 libraries, which should more closely mirror the human reference genome, the fraction of aligned reads not overlapping with any of the corresponding restriction sites in the reference genome was 0.80% and 0.96% for NlaIII and HindIII, respectively, indicating that these enzymes are extremely specific.

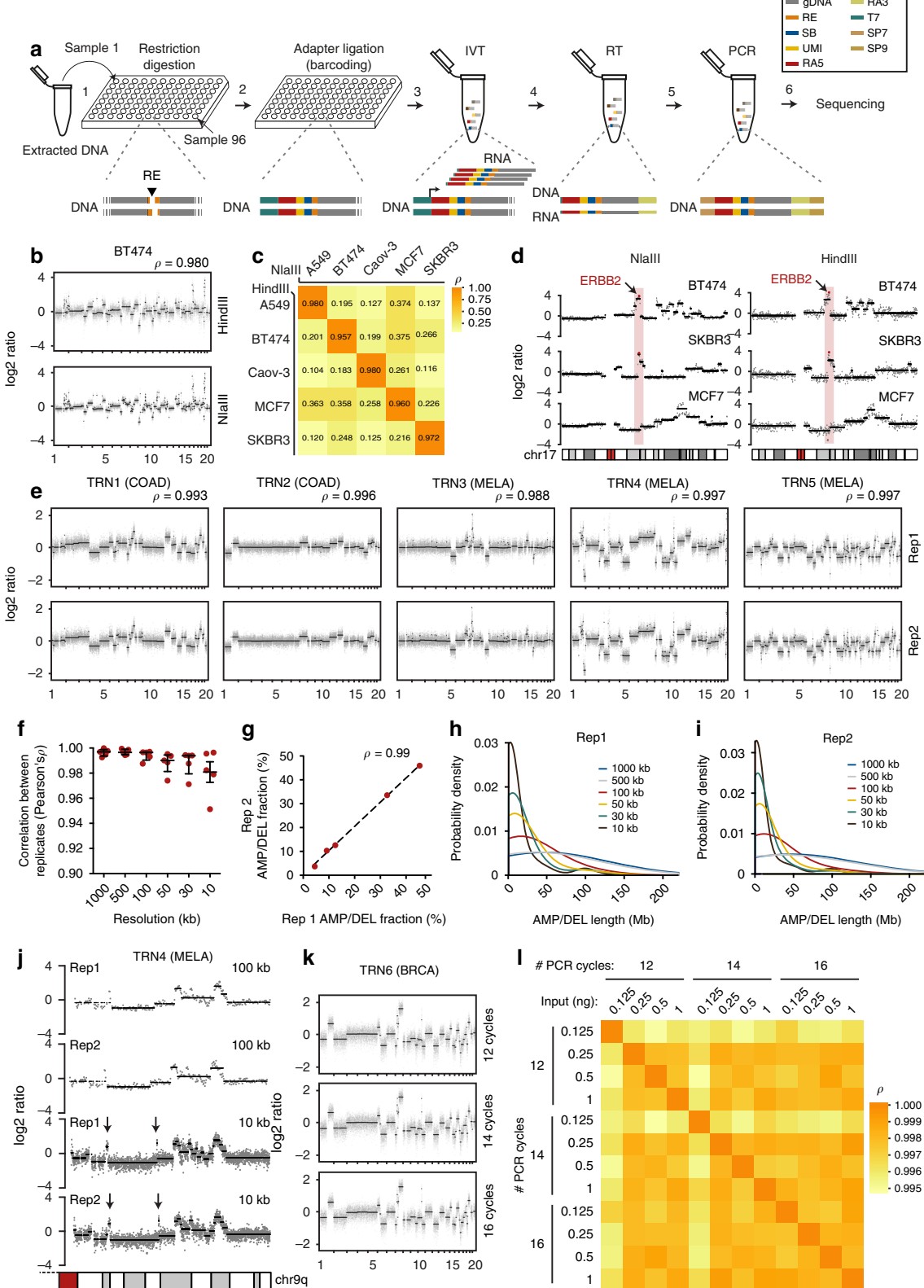

These results show that CUTseq is a valid method for preparing high-quality DNA libraries for sequencing on Illumina platforms.

**CUTseq reproducibility and sensitivity.** To evaluate the reproducibility of CUTseq, we first compared the DNA copy number profiles obtained with NlaIII and HindIII, at increasing resolutions ranging from 1 Mb up to 30 kb, for each of the cancer cell lines described above (Methods). The segmented copy number profiles were highly correlated between matched HindIII and NlaIII samples, at all the resolutions examined (Fig. 1b, c and Supplementary Fig. 3a, b). Each cell line showed a unique pattern of copy number alterations (CNAs), which were not correlated to the profiles of the other cell lines (Fig. 1c and Supplementary Fig. 3a, b), highlighting the specificity of CUTseq. Quantitative

**Fig. 1** CUTseq implementation and reproducibility. **a** CUTseq workflow. (1) RE, restriction enzyme. T7, T7 phage promoter. IVT, in vitro transcription. RA5, RA3, SP7, and SP9: Illumina's sequencing adapters. **b** BT474 cells copy number profiles (100 kb resolution). $\rho$, Pearson's correlation. **c** Pearson's correlation ($\rho$) between the copy number profiles (100 kb resolution) of five cancer cell lines digested with HindIII (rows) or NlaIII (columns). **d** Chr17 copy number profiles (NlaIII, 100 kb resolution) in two HER2-positive (SKBR3 and BT474) and one HER2-negative cell line (MCF7). *ERBB2/HER2* is highlighted in red. **e** Copy number profiles (NlaIII, 100 kb resolution) in five replicates (Rep) from FFPE tumor samples. COAD, colon adenocarcinoma. MELA, melanoma. $\rho$, Pearson's correlation. **f** Pearson's correlation ($\rho$) between the replicates shown in **e** at different resolutions. Each dot represents one pair of replicates. Error bars indicate the median and interquartile range. **g** Pearson's correlation ($\rho$) between the fraction of the genome (100 kb resolution) either amplified or deleted in the replicates (Rep) shown in **e**. Each dot represents one pair of replicates. Dashed line: linear regression. **h**, **i** Length of amplified (AMP) or deleted (DEL) genomic segments in Rep1 (**h**) and Rep2 (**i**) samples shown in **e**, at various resolutions. **j** Zoom-in view on chr9 q-arm in sample TRN4 shown in **e**. Arrows indicate focal amplifications detected only at 10 kb resolution in both replicates. Red: centromeric region. The p-arm is not shown. **k** Copy number profiles (NlaIII, 100 kb resolution) determined using 120 pg of gDNA extracted from one FFPE breast cancer (BRCA) sample and three different numbers of PCR cycles. **l** Pearson's correlation ($\rho$) between copy number profiles (100 kb resolution) determined using different amounts of gDNA extracted from the sample shown in **k**. In all the profiles, gray dots represent individual genomic windows, whereas black lines indicate segmented genomic intervals after circular binary segmentation[37]. The numbers below each box indicate chromosomes from chr1 (leftmost) to chr22 (rightmost). In all the cases, TRN refers to the ID of Turin samples, as shown in Supplementary Table 2. All the source data for this figure are provided as a Source Data file

analysis of the profiles revealed that, at comparable sequencing depth, the read count profiles fluctuated more in the case of HindIII-digested samples, which is expected based on the lower cutting frequency of this enzyme compared with NlaIII (Supplementary Figs. 1 and 3c, and Methods). In the case of IMR90, the DNA copy number profile was flat (Supplementary Fig. 3d), as expected for normal diploid cells. To confirm the specificity of CUTseq, we assessed the amplification status of the clinically relevant *ERBB2/HER2* oncogene on chromosome (chr) 17, which is amplified in BT474 and SKBR3 cells, but not in MCF7 cells, as previously shown[10,11]. Indeed, in BT474 and SKBR3 cells, but not in MCF7 cells, CUTseq detected a clear amplification of the *ERBB2* locus, both using HindIII and NlaIII (Fig. 1d). Thus, CUTseq is able to reproducibly detect cell type-specific copy number profiles using DNA extracted from cell lines.

We then assessed the reproducibility of CUTseq in FFPE samples. To this end, we first prepared two replicate libraries for each of five FFPE tumor samples, including two colon adenocarcinomas (COAD) and three melanomas (MELA) (Supplementary Table 2, Supplementary Data 2, and Methods). DNA copy number profiles were highly similar between replicates, across multiple resolutions (Fig. 1e, f and Supplementary Fig. 4). In line with this finding, the fraction of the genome that was detected as either amplified or deleted was highly correlated between corresponding replicates (Fig. 1g). By increasing the resolution, the distribution of the length of amplified and deleted genomic segments progressively shifted towards shorter lengths in a reproducible manner (Fig. 1h, i). Zooming-in on individual chromosomes revealed that the overall copy number profile was reproducible even at 10 kb resolution, whereas new features emerged reproducibly in both replicates at higher resolution (Fig. 1j), including focal amplifications and deletions, as well as more resolved complex patterns of alterations that could not be appreciated at lower resolutions (Supplementary Fig. 5a). High correlations between copy number profiles at multiple resolutions were also seen in CUTseq libraries prepared using increasing numbers of PCR cycles (Supplementary Fig. 5b, c and Methods), suggesting that extra amplification rounds do not significantly bias the copy number profiles. Furthermore, the correlation between replicates persisted by downsampling the number of reads (Supplementary Fig. 5d and Methods), demonstrating the ability of CUTseq to reproducibly detect CNAs even at relatively low sequencing depths.

Next, we investigated the sensitivity of CUTseq for picogram inputs of gDNA (125–500 pg), which most of commercially available kits cannot achieve (Supplementary Table 3). To this end, we prepared multiplexed libraries from gDNA extracted

from one breast cancer (BRCA) FFPE sample, by pooling into the same IVT reaction decreasing amounts of gDNA (1, 0.5, 0.25, and 0.125 ng) (Supplementary Table 2 and Methods). To further exclude the possibility of PCR biases, we prepared libraries using either 12, 14, or 16 PCR cycles. We then sequenced all the libraries and assessed DNA copy number profiles at various resolutions (Supplementary Data 2 and Methods). The segmented DNA copy number profiles remained extremely stable even for the 0.125 ng input and were highly correlated between each other, independently of the resolution and number of PCR cycles (Fig. 1k, l, Supplementary Fig. 6, and Supplementary Fig. 7a, b). Consistent with these observations, the overall fraction of the genome either amplified or deleted was relatively constant, independently of the gDNA input, number of PCR cycles, and resolution (Supplementary Fig. 7c), despite the fact that, as already observed in cell lines, the read count fluctuations progressively increased at higher resolutions and lower genome coverage (Supplementary Fig. 7d-f). Altogether, these results demonstrate that CUTseq is a reproducible and sensitive method that allows robust DNA copy number profiling across a broad range of resolutions, even for picogram amounts of gDNA extracted from FFPE samples.

**CUTseq benchmarking**. Next, we benchmarked CUTseq against standard methods of NGS library preparation. To do so, we used gDNA extracted from 10 FFPE samples representing four different tumor types, including four breast adenocarcinomas (BRCA), four COAD, two gastrointestinal stromal tumors (GIST), and two MELA samples (Supplementary Table 2 and Methods). For each sample, we constructed two libraries, one using CUTseq and the other using the commercially available library preparation kit, NEBNext® Ultra™ II (Methods). DNA copy number profiling at various resolutions (1 Mb up to 30 kb) revealed that the CUTseq and NEBNext profiles were strongly correlated, independently of the resolution (Fig. 2a, Supplementary Figs. 8 and 9, and Supplementary Fig. 10a). Consistent with this, the fraction of the genome that was detected as either amplified or deleted was highly correlated between matched samples (Fig. 2b and Supplementary Fig. 10b). Altogether, these results validate CUTseq as a sensitive and reliable method that can be used for DNA copy number profiling in FFPE samples, including low-input DNA specimens.

**Compatibility of CUTseq libraries with exome capture**. We then performed a proof-of-principle experiment to test whether CUTseq libraries are compatible with exome capture. To this end,

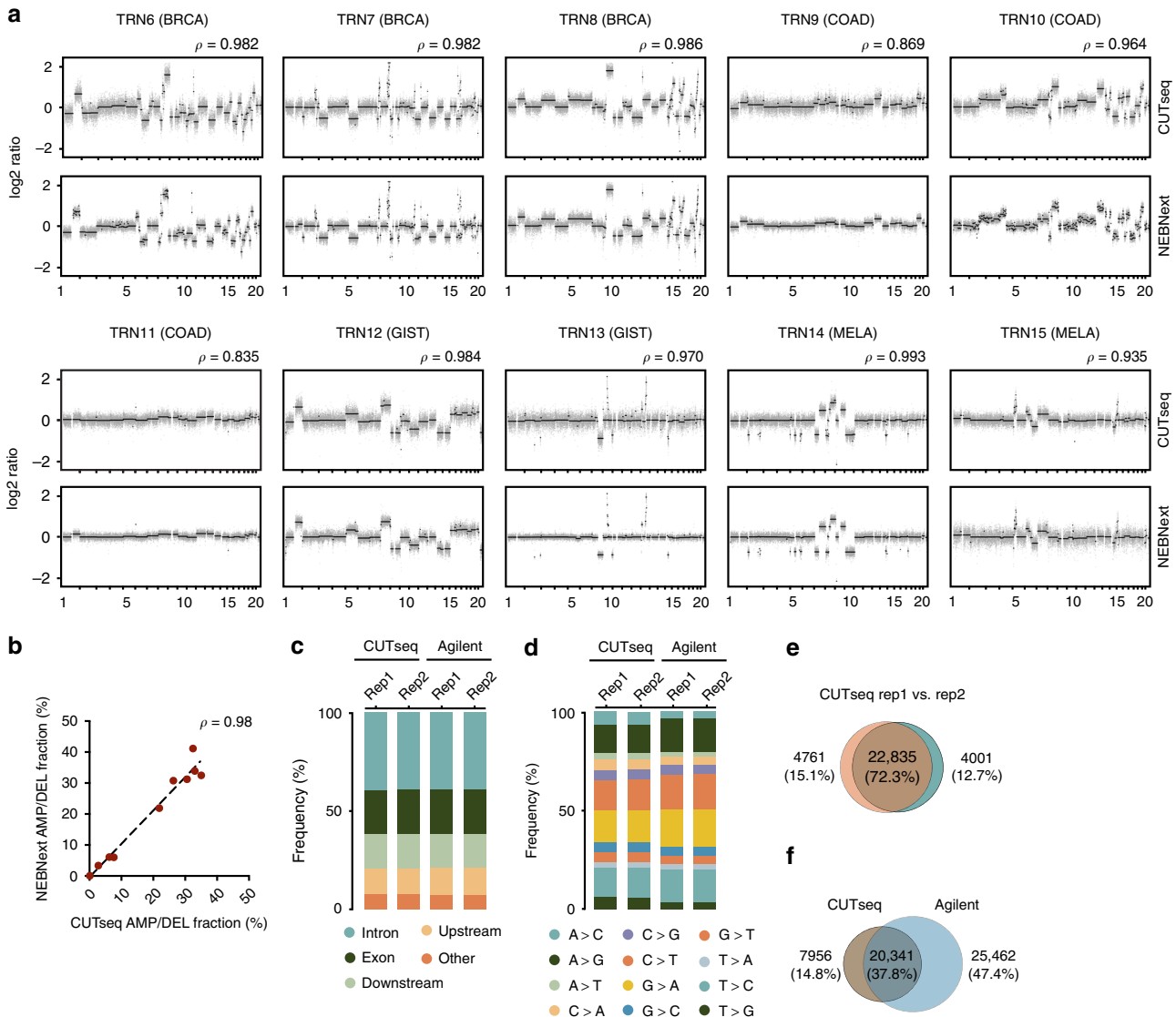

**Fig. 2** CUTseq validation. **a** Copy number profiles (NlaIII, 100 kb resolution) determined with CUTseq and NEBNext using gDNA extracted from ten different FPPE tumors. BRCA, breast cancer. COAD, colon adenocarcinoma. GIST, gastrointestinal stromal tumor. MELA, melanoma. $\rho$, Pearson's correlation between matched profiles. **b** Pearson's correlation ($\rho$) between the fraction of the genome (100 kb resolution) either amplified or deleted in each of the paired CUTseq and NEBNext samples shown in **a**. Each dot represents one pair of replicates. Dashed line: linear regression. **c** Partitioning of all the SNVs called in two replicate (Rep) exome capture experiments using SKBR3 cells gDNA and either CUTseq or a commercial kit for library preparation (Agilent), in multiple different annotated genomic regions. Up- and downstream indicate 5 kb windows before the start codons and after the stop codons of protein-coding genes, respectively. **d** Same as in **c**, but for different substitution types. **e** Overlap between the high-confidence SNVs (at least 50× coverage) called in the two CUTseq replicates shown in **c**, **d**. **f** Overlap between all the high-confidence SNVs identified by CUTseq vs. Agilent, after merging matched replicates shown in **c**, **d**. In both **e** and **f**, the percentages refer to the total number of SNVs in the union of the two sets. All the source data for this figure are provided as a Source Data file

we first prepared two replicate CUTseq libraries using gDNA extracted from SKBR3 cells and captured the exome using the SureSelect exome capture kit from Agilent Technologies. As a control, we prepared two replicate libraries from the same gDNA, but instead we used a commercial kit also from Agilent and captured them with the SureSelect kit (Supplementary Table 2, Supplementary Data 2, and Methods). SNV calling revealed that high-confidence SNVs (at least 50× coverage) were more concentrated around NlaIII recognition sites in CUTseq compared with Agilent samples (distance to closest NlaIII site: 77.08 ± 63.68 bp for CUTseq; 123.65 ± 142.65 bp for Agilent, mean ± SD) (Supplementary Fig. 10c), as indeed expected based on the fact that in CUTseq NlaIII was used to fragment the genome. The genomic distribution

and type of high-confidence SNVs were very similar between replicates and among CUTseq and Agilent samples (Fig. 2c, d). The high-confidence SNVs (72.3%) identified by CUTseq were detected in both replicates, whereas 37.8% of all the SNVs were shared between CUTseq and Agilent (Fig. 2e, f and Methods), even though the mean coverage per SNV was lower in CUTseq (Supplementary Fig. 10d), consistent with the fact that it is a reduced representation sequencing method. Similar results were obtained using gDNA extracted from two different FFPE tumor samples (Supplementary Fig. 10e, Supplementary Table 2, and Supplementary Data 2). Altogether, these results demonstrate that CUTseq libraries are compatible with standard exome capture and can thus be used for reduced representation exome sequencing.

**Multi-region tumor sequencing in FFPE tissue sections**. Next, we took advantage of the high sensitivity of CUTseq to assess intratumor heterogeneity of CNAs across multiple regions of individual breast cancer tissue sections. For this purpose, we retrieved 35 archival FFPE samples from 14 patients (age of specimens: 9–27 years), including primary tumors and one or more matched metastases previously profiled by whole exome sequencing[12] (Supplementary Table 2). For each tumor, we stained a 4 µm-thick section with hematoxylin–eosin and then extracted gDNA from a region L, ~7 mm² in diameter, which was confirmed by a pathologist to contain tumor cells (Fig. 3a and Methods). We split each region into half, to produce two technical replicates, L1 and L2 (Fig. 4a). In two cases, we also captured gDNA from multiple smaller regions S, ~3 mm² in diameter (Fig. 3a and Supplementary Fig. 11a). Accurate cell counting within 80 tumor regions of similar size in a different set of breast cancer samples revealed that, typically, such regions contain between 5000 and 25,000 cells (Supplementary Fig. 11b, c and Methods). Lastly, we extracted gDNA from the remaining material in the full tissue sections F, from which L and S regions were captured (Fig. 3a).

We separately barcoded the gDNA extracted from each region and then pooled multiple gDNAs into four libraries (Supplementary Data 2). In total, we barcoded 133 regions and sequenced each library aiming to obtain at least 200 K reads per region, which is sufficient for reliable copy number calling at 100 kb resolution. Indeed, the DNA copy number profiles of the matched L1 and L2 replicates appeared very similar (Fig. 3b), and the fraction of the genome that was detected as either amplified or deleted was highly correlated across replicates (Fig. 3c). Consistent with this observation, hierarchical clustering revealed that the profiles of matched L1 and L2 replicates always clustered together (Fig. 3d, e and Supplementary Fig. 12a), further highlighting the reproducibility of CUTseq. Typically, L regions also clustered together with the corresponding F regions (Supplementary Fig. 12a), suggesting that most of the tumor cells within a single tissue section harbor the same CNAs. These observations are in line with the notion that, in breast cancer, the majority of CNAs are acquired at an early stage during tumor evolution and therefore should be detectable across multiple tumor regions[13]. However, we also observed some exceptions. For example, in the metastasis-b of patient KI2, the L region showed a ~900 kb amplification on chr14q24, encompassing the *RAD51B* gene, which was reproducibly detected in both L1 and L2 replicates, but not in the full section (Fig. 3b, arrowhead). Similarly, two S regions in the primary tumor of patient KI14 clustered apart from all the other regions and showed numerous CNAs that were not detected in the corresponding F and L regions (Fig. 3b, e). These results highlight the importance of multi-region sequencing at high spatial resolution, to capture sub-clonal CNAs, which would otherwise go undetected when extracting gDNA from larger tissue areas.

Closer examination of the copy number profiles and hierarchical clustering trees also revealed that metastatic regions from the same tumor typically clustered together, and apart from the regions of the corresponding primary lesion (Fig. 3b-e and Supplementary Fig. 12a). Moreover, among all the regions with detectable CNAs, the metastatic regions had a significantly higher burden of amplifications and deletions compared with the primary tumor regions (P-value = 0.006, Mann–Whitney test, two-tailed) (Supplementary Fig. 12b). These results are in agreement with the findings of a recent study on a larger sample cohort, according to which breast cancer distant metastases typically show a different, although phylogenetically related, mutational landscape compared with the corresponding primary tumors, as a result of ongoing genome instability and tumor evolution[14].

Finally, we checked how many of the 712 cancer-associated genes in the COSMIC database[15] are affected by CNAs in different tumor regions. Two hundred and forty-one of the 712 genes (33.8%) were amplified, whereas 261 genes (36.6%) were deleted in one or more tumor sites, regions, or patients in our cohort. The top-three amplified genes were *MYC*, *NDRG1*, and *RAD21*, whereas *KMTA*, *PAFAH1B2*, and *POU2AF1* were the three most frequently deleted genes (Fig. 3f, g and Methods). Hierarchical clustering revealed at least two major groups of samples: one group harboring amplifications and deletions in a large subset of COSMIC genes; and the other group predominantly characterized by amplifications in a smaller subset of COSMIC genes, including many genes that are recurrently affected by CNAs in breast cancer[16], such as *MYC*, *ERBB2*, *CCND1*, *MDM2*, and *PIK3CA* (Fig. 3h and Methods). Among frequently amplified genes, *MYC* and *ERBB2* were amplified in 7 and 8 out of 14 patients, respectively (50% and 57%), whereas, among frequently deleted genes, the classical onco-suppressor *TP53* gene was deleted in 4 out of 14 patients (28.6%). Five primary tumors in which CUTseq detected *HER2* amplification (KI2, 4, 10, 11, 12) were also HER2-positive based on immunohistochemistry (Supplementary Table 2), further validating our method. In one case (KI7), CUTseq detected HER2 amplification only in the metastasis, but not in the corresponding primary tumor (Fig. 3b, arrowhead), in line with recent observations that some breast cancers classified as HER2-negative might actually express HER2 at distant metastatic sites[17]. Overall, these results demonstrate that CUTseq is a robust and sensitive method that can be used to profile, at high spatial resolution, DNA CNAs across multiple regions in clinically relevant tumor samples, thus providing valuable insights into intratumor genetic heterogeneity.

**High-throughput CUTseq**. Lastly, we aimed to streamline the preparation of highly multiplexed CUTseq libraries. To reduce the assay cost and turnaround time, we developed a workflow that takes only ~8 h from DNA digestion to ready-to-sequence libraries (Fig. 4a, and Methods). To reduce reagent volumes, and therefore costs, we used a contactless liquid-dispensing robot, which allows performing digestion and ligation reactions in nanoliter volumes (Fig. 4a). As a proof-of-principle, we prepared a multiplexed library by digesting and differentially barcoding 96 replicate samples of HeLa cells gDNA inside a 96-well plate (5 ng per well) and then pooled all the samples into a single IVT reaction (Methods). We sequenced all the samples shallowly on NextSeq 500, obtaining 88 out of 96 replicates (91.7%) with at least 100 K usable reads (Fig. 4b and Methods). Notably, the sequencing error rate was very low and typically comprised between 1.5% and 1.7% (median: 1.62%; interquartile range: 1.58%–1.68%) (Fig. 4c), highlighting the precision of CUTseq, even when quick digestion and ligation are performed in nanoliter volumes. In the 88 replicates with at least 100 K usable reads, the DNA copy number profiles appeared highly similar (Fig. 3d) and were strongly correlated between each other (Fig. 4d, e and Methods). In line with this, the fraction of the genome that was detected as either amplified or deleted was very homogenous across replicates (Fig. 4f). Importantly, the cumulative cost of preparing libraries for a large number of samples is substantially lower for CUTseq compared with available commercial kits, independently of the use of a nanoliter dispensing device (Supplementary Note 1). These results demonstrate that high-throughput CUTseq is a cost-efficient method for sequencing multiple samples in parallel, including low-input gDNA samples.

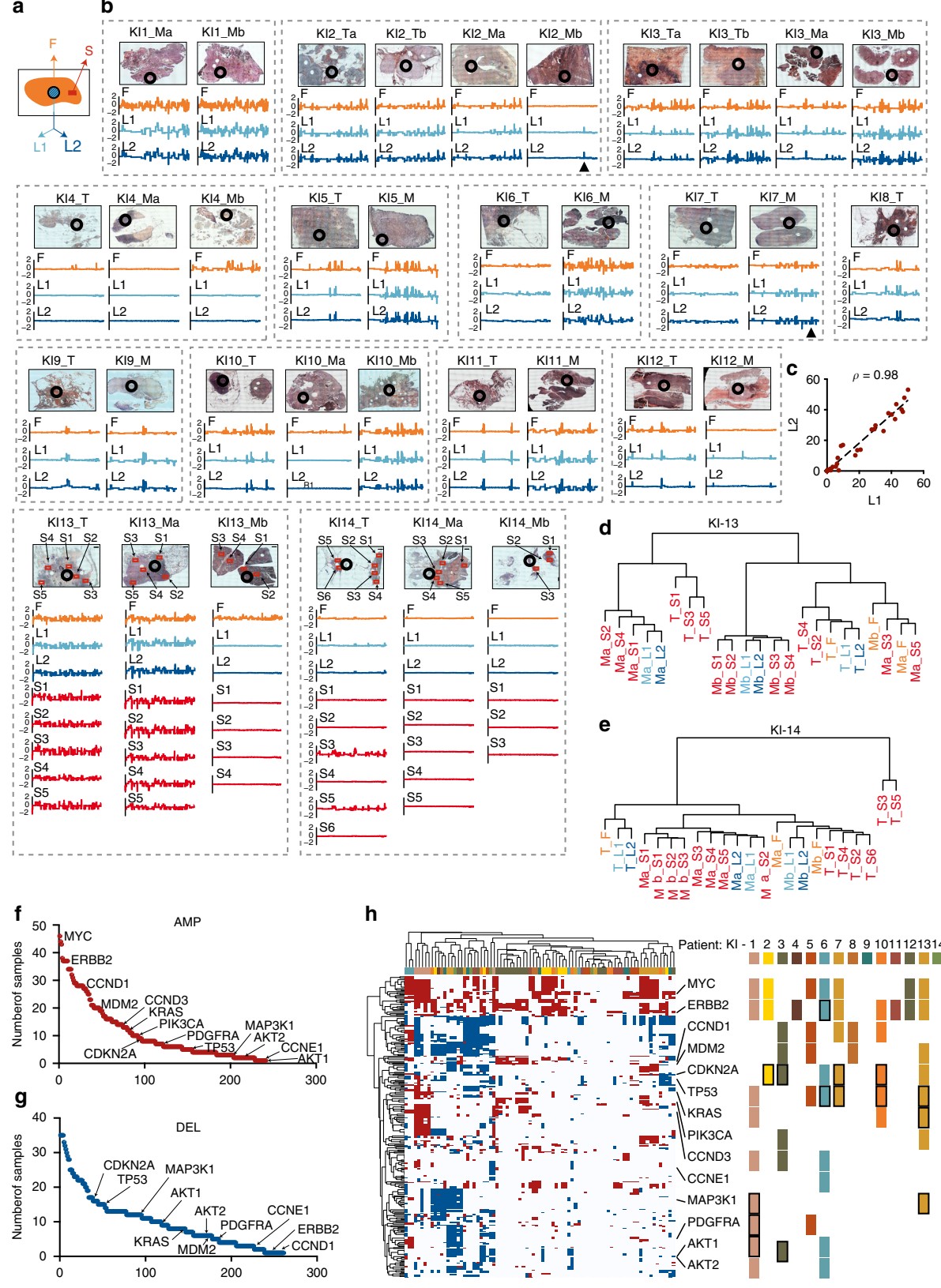

## Discussion

We have developed a streamlined method for gDNA barcoding and amplification, which enables the generation of multiplexed DNA sequencing libraries from both fixed and non-fixed cell and tissue samples, including single FFPE tissue sections or small regions thereof. The key advantage of CUTseq compared with standard methods of NGS library preparation is that each sample gets barcoded upfront, instead of at the end of the library preparation workflow, which allows multiple samples to be pooled together into the same library. This is possible, thanks to the combination of two widely available molecular biology tools: (i) type-II restriction enzymes that produce stereotypic DNA

**Fig. 3** Multi-region copy number profiling in FFPE breast cancer tissue sections. **a** Scheme of regions within individual FFPE breast cancer sections from which gDNA was extracted. S, small regions of ~3 mm$^2$. L, large regions of ~7 mm$^2$. For each L region, gDNA was split in two technical replicates, L1 and L2. F, remaining tissue in the section. **b** Scans (×10 magnification) of 35 hematoxylin–eosin-stained tissue sections from primary (T) and metastatic (M) breast cancers, and corresponding copy number profiles (100 kb resolution), for F, L, and S regions. Black circles: L region from which L1 and L2 replicates were obtained. Black arrowheads: amplification of the *RAD51B* gene in patient KI2 and of the *HER2* gene in patient KI7. In all the profiles, chr1 is on the left and chr22 on the right. **c** Pearson's correlation (*ρ*) between the fraction of the genome either amplified or deleted in all the L1–L2 replicates shown in **b**. **d** Hierarchical clustering of copy number profiles for F, L, and R regions, in patient KI13. **e** Same as **d**, but for patient KI14. **f** Ranking of 712 cancer-associated genes in COSMIC[15] based on the number of samples in which they were found amplified (AMP). Gene names refer to a subset of 31 COSMIC genes that were found to be frequently amplified or deleted in 560 breast cancers[16]. **g** Same as **f**, but for genes deleted (DEL) in the samples shown in **b**. **h** Hierarchical clustering of 712 COSMIC genes (rows) based on their amplification (red) or deletion (blue) status in the 133 samples shown in **b** (columns). Gene names indicate 14 out of 31 genes that were previously found to be either amplified or deleted in breast cancer[16]. For each gene, the rectangles on the right indicate whether it is amplified (no boundary) or deleted (black boundary) in at least one sample (F, L, R) in the corresponding KI patient. In all cases, KI refers to the ID of samples from Karolinska Institute, as described in Supplementary Table 2. All the source data for this figure are provided as a Source Data file

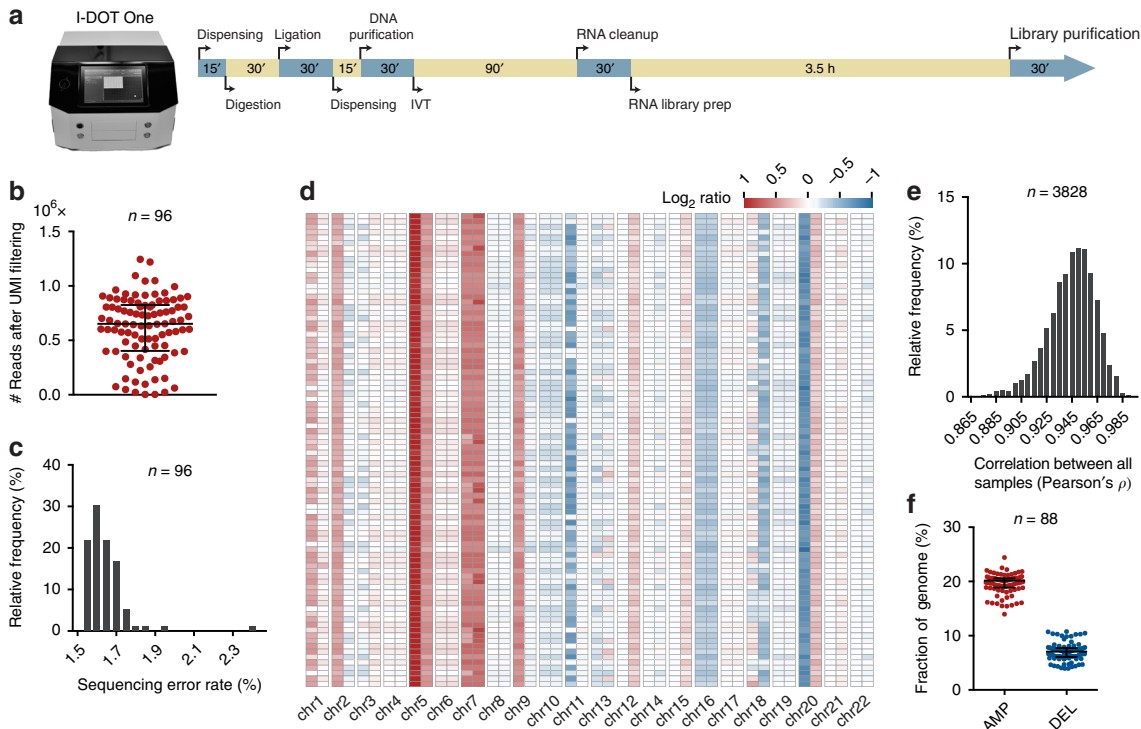

**Fig. 4** High-throughput CUTseq. **a** Front picture of the I-DOT One MC, low-volume non-contact dispensing device (Dispendix) that was used in this study, and timeline for high-throughput CUTseq library preparation. IVT, in vitro transcription. The total workflow takes ~8 h for a single person to prepare 1–2 libraries, each containing up to 96 samples. The dispensing step can be done either manually or using a liquid handling device such as I-DOT One. **b** Number of usable reads (after alignment and PCR duplicates removal) per sample, in one multiplexed CUTseq library prepared from 96 replicate samples (*n*) of HeLa cells gDNA (5 ng), using I-DOT One. **c** Distribution of the sequencing error rates in the 96 replicates (*n*) shown in **b**. **d** Copy number profiles (1 Mb resolution, averaged at arm level for visualization) of 88 replicates shown in **b** that yielded at least 300 K usable reads. The remaining eight samples were not included, as the number of usable reads was insufficient to perform reliable copy number calling. **e** Distribution of all possible (*n*) pairwise Pearson's correlations between the copy number profiles shown in **d**. **f** Fractions of the genome either amplified (AMP) or deleted (AMP) in the 88 replicates (*n*) shown in **d**. Each dot represents one sample. Error bars indicate the median and interquartile range. All the source data for this figure are provided as a Source Data file

overhangs, to which complementary adapters can be immediately ligated without the need for end-repair, unlike what is done in most of conventional NGS library preparation methods (Supplementary Table 3); and (ii) IVT, which allows pooling together and co-amplifying multiple samples in the same reaction. Another advantage is the incorporation of UMIs[9] at the site of CUTseq adapter ligation, which allows post-sequencing removal of PCR duplicates and single-molecule counting, without having to perform paired-end sequencing. Thanks to all these features, multiple samples can be merged into a single library and sequenced together without the need to prepare and quantify

multiple libraries, which drastically reduces the overall cost per sample, as we demonstrate in the Supplementary Note 1. Importantly, multiplexing is not only helpful to reduce costs, but is also particularly advantageous when dealing with low-input samples for which it is challenging to prepare single-sample libraries using standard technology. As we have shown here, by pooling multiple low-input samples into the same CUTseq library, we were able to obtain very reliable DNA copy number information at kilobase resolution, even for samples of only 120 pg of FFPE gDNA, which most of the existing commercial kits for NGS library preparation cannot do (see Supplementary Table 3).

One distinguishing feature of CUTseq compared with conventional NGS library preparation methods is that it uses restriction enzymes instead of random genome fragmentation, thus providing a reduced representation of the genome. The choice of the restriction enzyme depends on the cutting frequency along the genome as well as on the desired resolution. As shown in Supplementary Figs. 3c and 7d-f, the fluctuation of read counts around the segmented genomic profiles is influenced by various parameters, including sequencing depth (genome coverage), binning size (resolution), and the cutting frequency of the restriction enzyme in use. In general, at comparable sequencing depths, profiles generated with a four-base cutter appear less noisy than profiles obtained with a six-base cutter, especially at high resolutions. However, it is critical to note that, despite increasing noise levels in the raw read count profiles, the segmented profiles are extremely stable even at high resolution and picogram DNA inputs. As a rule of thumb, we recommend using a four-base cutter such as NlaIII when high resolution is desired (<50 kb), otherwise a less expensive six-base cutter such as HindIII.

Although reduced genome representation does not prevent accurate DNA copy number calling at high resolution, as we have shown here, the same feature inherently limits the ability to detect SNVs at any position in the genome. However, as we have also demonstrated in this study, CUTseq is able to reproducibly detect a considerable fraction of high-confidence SNVs detected by a standard exome capture method and, as such, it can be used for reduced representation exome sequencing. One application of reduced representation exome sequencing would be in multi-region tumor sequencing, to detect a lower number of high-confidence SNV events, but from many more regions than currently possible, at comparable sequencing costs. This would significantly improve the ability to reconstruct a tumor's phylogeny, by comparing CNA and SNV profiles from many regions in the same tumor. Even though in this study we have used single-end sequencing and short reads, combining a frequent cutter with paired-end sequencing and long reads should, in principle, allow for higher exome coverage. Furthermore, using a cocktail of different enzymes could also increase the exome coverage. For example, we found that over 15,000 recurrent mutations in 127 genes frequently mutated in 12 major cancer types[18] are <500 bp away from the closest NlaIII recognition site (Supplementary Fig. 13a). In line with this, the mean number of NlaIII recognition sites in the exons of cancer-associated genes listed in the COSMIC database[15] is 4.3 $kb^{-1}$ (median = 4.2 $kb^{-1}$, SD = 1.3 $kb^{-1}$) (Supplementary Fig. 13b), which means that most of the cancer mutations are, at least in principle, detectable with CUTseq. Thus, CUTseq is a valuable method that expands the existing toolkit for studying cancer genomes.

Compared with other reduced representation genome-sequencing methods, such as the RAD-seq method[19], which is widely used in population genetics and ecology[20], CUTseq requires only one, and not two, ligation events, to barcode gDNA and amplify it by IVT. This means that, for a given gDNA fragment, the probability of getting properly ligated and barcoded is higher for CUTseq compared with RAD-seq. This is particularly advantageous, especially in cases in which the starting material is very little, as in the case of gDNA extracted from small regions within individual FFPE tissue sections. Furthermore, although in RAD-seq DNA libraries are typically prepared from individual samples[20], the high-throughput CUTseq workflow described here offers a streamlined and cost-effective solution for analyzing hundreds of specimens in parallel, and thus could be very useful in ecology and population genomics applications.

As a proof-of-principle, we have shown an application of CUTseq to assess DNA copy number profiles across multiple regions inside individual FFPE sections of primary and metastatic breast cancer lesions, after assessing them by conventional histology. Our results demonstrate that, by differentially barcoding the gDNA extracted from multiple small regions within the same tissue section, it is possible to assess the extent of genetic intra-tumor heterogeneity and pinpoint alterations that would otherwise go undetected by sequencing gDNA extracted from larger tissue areas. Importantly, the cost of preparing a multiplexed library from multiple regions within a single FFPE tissue section using CUTseq is very similar to the cost of preparing a single library from the entire tissue section using a standard method, such as one of those shown in Supplementary Table 3. Thus, CUTseq could be applied in routine diagnostics to assess DNA CNAs and intratumor genetic heterogeneity directly in the tissue sections that have been used for pathological diagnosis.

Another possible area of application of high-throughput CUTseq outside tumor sequencing is copy number profiling in the frame of genetic screens and cell line authentication efforts. For example, it has been recently reported that CRISPR nucleases, which are widely used for genome editing, can cause unwanted large deletions and complex rearrangements[21,22]. In this context, high-throughput CUTseq could be used to screen whether multiple CRISPR nucleases and small-guide RNA constructs induce large-scale CNAs. Similarly, CUTseq could be applied for cell line authentication and monitoring genome stability in cultured cells. Recent international efforts to identify cross-contamination among popular cell lines, such as the International Cell Line Authentication Committee (ICLAC)[23], would greatly benefit from CUTseq to cost-efficiently profile the genomic landscape and stability of hundreds of cell lines in public repositories such as the American Tissue Culture Collection and the Coriell Repository. In conclusion, CUTseq is a versatile, quantitative, and streamlined method for reduced representation genome sequencing with broad applications in both research and diagnostics.

## Methods

**Cell lines**. We purchased the following cell lines from ATCC: IMR90 (catalog number CCL-186), BT474 (catalog number HTB-20), A549 (catalog number CCL-185), MCF7 (catalog number HTB-22), HeLa (catalog number CCL-2), Caov3 (catalog number HTB-75), and SKBR3 (catalog number HTB-30). None of these cell lines is included in the ICLAC database of commonly misidentified cell lines. We cultured IMR90 cells in MEM (Gibco, catalog number 10370021) supplemented with 10% non-heat-inactivated fetal bovine serum (FBS; Gibco, catalog number 16000044), 2 mM L-glutamine (Sigma, catalog number 59202C), and 1% non-essential amino acids (Gibco, catalog number 11140035); A549 cells in RPMI 1640 (Sigma, catalog number R8758) supplemented with 10% heat-inactivated FBS (Sigma, catalog number F9665); BT474, MCF7, HeLa, and Caov3 cells in Dulbecco's modified Eagle's medium (Sigma, catalog number D6429) supplemented with 10% heat-inactivated FBS (Sigma, catalog number F9665); and SKBR3 in McCoy's 5A (Sigma, catalog number M9309) supplemented with 10% heat-inactivated FBS (Sigma, catalog number F9665). We incubated cells at 37 °C in 5% $CO_2$ air. We tested all the cell lines for mycoplasma contamination using the MycoAlert Mycoplasma Detection Kit (Lonza, catalog number LT07-118), but we did not authenticate them.

**FFPE samples**. TRN samples (see Supplementary Table 2). We retrieved 31 FFPE tumor samples of different origin (GIST, COAD, BRCA, and MELA) at the Pathology Unit of IRCC Candiolo, Italy, in the frame of a prospective study approved by the "Istituto di Candiolo FPO-IRCCS" Ethical Committee for the identification of molecular profiles conferring resistance to selected target therapies in oncological patients ("Profiling" # 001-IRCC-00IIS-10).

KI samples (see Supplementary Table 2). We collected one FFPE tissue section (4 μm-thick) per lesion, from both the primary tumor and 1 or more distant metastases that occurred in 14 female patients. We identified the patients by searching the KI electronic medical records. A board-certified surgical pathologist at Karolinska Institutet diagnosed the lesions as metastatic breast cancer. This study was approved by the local ethical committee at Karolinska Institutet under permission number 2013/1273-31/4 with amendments 2013/1739-32 and 2014/707-32. We deparaffinized FFPE tissue sections in xylene and stained them with hematoxylin for 8 min. Afterwards, we rinsed the sections with running tap water for 5 min and then immersed them in eosin for 2 min. The sections were ready to use after dehydration in ethanol[16].

**Tissue imaging and automated cell counting**. We stained all the 35 FFPE breast cancer sections that were used for multi-region tumor sequencing (see Fig. 3) using hematoxylin–eosin. Before staining, we deparaffinized all the sections in xylene (Honeywell, catalog number 534056) and rehydrated them using an alcohol scale. We scanned each tissue section using an Eclipse Ti inverted wide-field fluorescence microscope (Nikon, Japan) in phase-contrast mode with a ×10 objective. To count the number of cells in tissue regions of size comparable to the regions from which gDNA was captured, we stained an independent set of 16 FFPE tissue sections from 16 different breast cancers (Supplementary Table 2) with 1 ng/µl Hoechst 33342 (Thermo Fisher, catalog number 62249) in 1× phosphate-buffered saline (PBS), for 15 min at 30 °C. We then scanned a 1 × 1 cm region in each section using an Eclipse Ti2 inverted epifluorescence microscope (Nikon, Japan) at ×40 magnification. To automatically segment the cell nuclei, we used the Ilastik[24] open-source pixel classifier software, by training the software on a single scan. We converted the segmentation masks obtained with Ilastik to 8-bit images and binarized them using FIJI[25]. Afterwards, we applied the following functions available in FIJI by combining them into a single macro, which is provided as Supplementary Software: first "open", to remove isolated pixels; then "fill holes" and "watershed segmentation", to further improve the segmentation obtained with Ilastik; lastly, "analyze particles" excluding objects smaller than 100 square pixels, to count cells. We counted cells in five 1.7 × 1.5 mm regions in each tissue section, by selecting the regions so that they overlap with tumor-dense areas annotated in the same section by a certified pathologist.

**gDNA extraction**. *Cultured cells*. We first trypsinized the cells with 0.25% (w/v) trypsin-EDTA (Ambion, catalog number AM9261) when they reached confluency and resuspended them in fresh culturing medium. After centrifuging, we resuspended the cell pellet and washed it twice in 1× PBS (Ambion, catalog number AM9625). Lastly, we lysed the cell pellet using a buffer containing 10 mM Tris-HCl/100 mM NaCl/50 mM EDTA/1% SDS/19 mg/ml Proteinase K (NEB, catalog number P8107S), pH 7.5, and incubated the solution overnight at 55 °C on a thermomixer, shaking at 800 r.p.m. The following day, we purified gDNA using a standard phenol–chloroform extraction protocol. We quantified the gDNA using the Qubit 2.0 Fluorimeter and the High Sensitivity DNA Kit (Agilent, catalog number 5067–4626). We note that gDNA extracted with silica-based kits is also perfectly compatible with the subsequent steps of CUTseq. More details are provided in the step-by-step protocol available in the Supplementary Information and at Protocol Exchange (https://doi.org/10.21203/rs.2.1742/v1).

*TRN samples* (see Supplementary Table 2). We extracted 200 ng of gDNA from five representative 10 µm-thick sections with >50% tumor cells after manual dissection, using the QIAamp DNA FFPE Tissue Kit (Qiagen, catalog number 56404) according to the manufacturer's protocol. We quantified the gDNA using the Qubit 2.0 Fluorimeter and the High Sensitivity DNA Kit (Agilent, catalog number 5067–4626).

*KI samples* (see Supplementary Table 2). To extract gDNA from multiple regions in individual FFPE sections, we first used the PinPoint Slide DNA Isolation System™ (ZymoResearch, catalog number D3001) to capture selected regions (see Fig. 3). Afterwards, we captured all the remaining tissue section also using Pinpoint. After air drying the samples for at least 30 min at room temperature, we used sterile disposable insulin needles to pick up the dried gDNA and transfer it into a DNA LoBind tube (Sigma, catalog number Z666548). We then resuspended and lysed the tissue in the same buffer used for cell lines, and purified gDNA using a standard phenol–chloroform extraction protocol.

**CUTseq**. A step-by-step protocol is available in the Supplementary Information as well as at Protocol Exchange (https://doi.org/10.21203/rs.2.1742/v1). To prepare CUTseq adapters, we purchased the oligonucleotides listed in the Supplementary Data 1 as standard desalted oligos from Integrated DNA Technologies. UMIs were generated by random incorporation of the four standard dNTPs using the "Machine mixing" option. We first diluted the oligos at 10 µM in nuclease-free water. We phosphorylated the upper oligos with 20 U of T4 Polynucleotide Kinase (NEB, catalog number M0201) in a final volume of 90 µl, by incubating for 1 h at 37 °C. Afterwards, we added an equal volume of the corresponding antisense oligos pre-diluted at 10 µM in nuclease-free water and incubated the solution for 5 min at 95 °C, followed by cooling down to 25 °C over a period of 45 min in a PCR thermocycler. We digested purified gDNA with 20 U of HindIII (NEB, catalog number R3104) or NlaIII (NEB, catalog number R0125) enzyme in a final volume of 10 µl, by incubating for 14 h at 37 °C. Afterwards, we ligated HindIII or NlaIII cut sites with CUTseq adapters carrying the complementary staggered end, using 1000 U of T4 ligase (Thermo Fisher Scientific, catalog number EL0014) in a final volume of 30 µl, by incubating for 18 h at 16 °C. After ligation, we purified gDNA with 3.7 µl/100 µl glycogen (Sigma, catalog number 10901393001)/sodium acetate, pH 5.5 (Life Technologies, catalog number AM9740)/ice-cold ethanol (VWR, catalog number 20816.367). For IVT, we used the MEGAscript® T7 Transcription kit (Thermo, catalog number AM1334-5) following the manufacturer's protocol. In case of gDNA extracted from small regions of single tissue sections––for which we could not measure the concentration––we pooled together gDNA extracted from up to 48 regions in a single IVT reaction of 20 µl and incubated for 14 h at 37 °C. In order to prevent RNA degradation, we added 20 U of RNaseOUT™ Recombinant Ribonuclease Inhibitor (Invitrogen, catalog number 10777-019) per 20 µl of IVT

reaction. After IVT, we purified the amplified RNA with Agencourt RNAClean XP beads (Beckman Coulter, catalog number A63987) following the manufacturer's instructions. Lastly, we prepared sequencing libraries from the amplified RNA, using the TruSeq Small RNA Library Preparation kit (Illumina, catalog number RS-200-0012/RS-200-0024). For CUTseq validation, we prepared libraries with the NEBNext® Ultra™ II DNA Library Prep Kit for Illumina® (NEB, catalog number E7645/E7103) and NEBNext adaptors (NEB, catalog number E7350), following the manufacturer's instructions. We analyzed the size distribution and concentration of all the libraries using a Bioanalyzer 2100 (Agilent Technologies, catalog number G2943CA) and the High Sensitivity DNA kit (Agilent Technologies, catalog number 5067–4626), and sequenced them on a NextSeq 500 system (Illumina) using the NextSeq 500/550 High Output v2 kit (75 cycles) (Illumina, catalog number FC-404-2005).

**CUTseq with serial gDNA dilutions and different PCR cycles**. To test the reproducibility of CUTseq for different input amounts of the same gDNA and effects on the DNA copy number profiles with different PCR cycles, we extracted gDNA from a single FFPE section of one colon cancer (TRN1) and one breast cancer (TRN6, see Supplementary Table 2) using the procedure described above for TRN samples. We measured the concentration of gDNA with the Qubit 2.0 Fluorimeter and the High Sensitivity DNA Kit (Agilent, catalog number 5067–4626). For TRN1, we prepared a single-sample CUTseq library using 200 ng of gDNA and 10 PCR cycles. Afterwards, we prepared four libraries, by using 1 µl per sample of the purified library and subjected them to two, four, six, and eight extra PCR cycles. For TRN6, we prepared three multiplexed CUTseq libraries by pooling barcoded decreasing input amounts of the same extracted gDNA: 1.0, 0.5, 0.25, and 0.125 ng using either 12, 14, or 16 PCR cycles. We analyzed all the libraries on Bioanalyzer and sequenced them using the NextSeq 500/550 High Output v2 kit (75 cycles) (Illumina, catalog number FC-404-2005). A list of recommended PCR cycles depending on the amount of input gDNA is provided in Supplementary Table 4.

**Exome capture**. To test whether CUTseq can be used for exome sequencing, we extracted gDNA from SKBR3 cells and two FFPE breast samples (TRN16 and TRN17, see Supplementary Table 2). For SKBR3 cell sample, we prepared two CUTseq libraries as well as two reference libraries using the SureSelect XT HS Kit (Agilent Technologies, catalog number G9704A). For the two FFPE breast samples, we prepared one CUTseq library and one reference library for each. The amount of gDNA input was 50 ng in all the cases. We then performed exome capture using the SureSelect XT HS Target Enrichment kit and SureSelect Human All Exon v6 baits (Agilent Technologies, catalog number G9704K) on all libraries following the manufacturer's protocol. We analyzed all the captured libraries on Bioanalyzer and sequenced them using the NextSeq 500/550 High Output v2.5 kit 300 cycles (Illumina, catalog number 20024908).

**High-throughput CUTseq**. To streamline the preparation of multiplexed libraries from low-input samples, we adapted the CUTseq workflow to perform the digestion and ligation steps in multi-well plates using a low-volume non-contact liquid-dispensing system (I-DOT One MC from Dispendix GmbH, Germany). Briefly, we first manually dispensed 5 µl of Vapor-Lock (Qiagen, catalog number 981611) per well in 96 wells of a 384-well plate. We then used I-DOT One to dispense first 5 ng diluted in 350 nl of gDNA extracted from HeLa cells, followed by 100 nl of 20 U/µl of HindIII (NEB, catalog number R3104) and 50 nl of CutSmart buffer (NEB, catalog number R3104), in 96 of the 384 wells. After incubating the plate at 37 °C for 30 min, we used I-DOT again to dispense 300 nl per well of CUTseq adapter at 33 nM, using a differently barcoded adapter for each well, followed by 200 nl of T4 rapid DNA ligase (Thermo Fisher Scientific, catalog number K1423), 300 nl of T4 ligase buffer 5× (Thermo Fisher Scientific, catalog number K1423), 120 nl of ATP at 10 µM (NEB, catalog number P0756L), 30 nl of 50 mg/ml bovine serum albumin (Thermo Fisher Scientific, catalog number AM2616), and 50 nl of nuclease-free water (Thermo Fisher Scientific, catalog number 4387936). We incubated the plate at 25 °C for 30 min and then pooled all the contents of the 96 wells used into 1 tube, before proceeding to IVT and library preparation following the standard CUTseq protocol. A list of reagents and volumes dispensed using the I-DOT One system is provided in Supplementary Table 5.

**Sequencing data processing**. We demultiplexed the raw sequence reads based on index sequences using the BaseSpace® Sequence Hub cloud service of Illumina. Afterwards, FASTQ files were generated and downloaded from BaseSpace. To process FASTQ files, we developed a custom pipeline that is freely available on GitHub at this address: https://github.com/garner1/cutseq. Briefly, we retained reads containing the expected prefix consisting of 8 nt UMI and 8 nt sample barcode using SAMtools (v1.9)[26] and scan_for_matches[27], allowing at most two mismatches in the sample barcode. We then clipped off the prefixes and aligned the trimmed reads to the GRCh37/hg19 reference genome using BWA-MEM[28] with default parameters. We retained reads that aligned with a quality score ≥ 30. Lastly, we identified and removed PCR duplicates using the Python umi_tools[29] package,

with default parameters. We excluded reads aligned to sex chromosomes, to consider always comparable genomic support regions.

**Estimation of sequencing error rates**. We estimated the sequencing error rate in all our sequencing experiments, by using the information contained in the Cigar string of the generated BAM files and the open-source Alfred software[30]. The sequence error rate is calculated by dividing the total number of deletions, insertions, and mismatches (characters D, I, and M with mismatch to reference in the Cigar string in the BAM files) over the total number of aligned bases (character M in the Cigar string).

**Copy number calling and analysis**. To determine DNA copy number levels, we used the R package QDNAseq[31], which is optimized for FFPE samples. The threshold used for calling amplifications and deletions was $log_2 \frac{2.5}{2}$ and $log_2 \frac{1.5}{2}$, respectively. We binned the genomes in non-overlapping windows of constant size (1 Mb, 500 kb, 100 kb, 50 kb, or 30 kb) and plotted genome-wide copy number profiles using custom scripts in R. We plotted chromosome-specific profiles together with the corresponding chromosome ideogram using the R package karyoploteR[32]. To quantify the similarity of copy number profiles across different samples, we computed the Pearson's pairwise correlation of the log ratio values in corresponding genomic windows using custom scripts in R. To quantify the fluctuation of the CUTseq signal, we computed the mean and SD of the absolute difference between the non-segmented and segmented log2 ratio values inside each genomic window. To compare copy number profiles obtained at different sequencing depths, we down-sampled the SAM file using SAMtools (v1.9)[26]. To determine the aneuploid genome fraction, we calculated the percentage of 100 kb genomic windows that are called either amplified or deleted using custom scripts in R. To identify cancer-associated genes that were either amplified or deleted in the KI samples, we downloaded a list of genes frequently mutated in cancers from the COSMIC database[15]. We then clustered the samples by calculating the Euclidean distances between the state of each COSMIC gene (amplified $= 1$, deleted $= -1$, or neutral $= 0$) in all the samples, followed by complete linkage clustering. Furthermore, to examine COSMIC genes related to breast cancer, we used a list of genes that are frequently affected by CNAs in breast cancer, which is included in the Supplementary Table 14 of ref. [16]. To cluster the copy number profiles of different regions in the FFPE sections of KI samples, we used the R function hclust with default parameters and complete linkage clustering.

**Single-nucleotide variants calling**. To analyze the sequencing data obtained from exome capture, we performed SNV calling using the bcbio-nextgen tool (v1.1.4a) (https://github.com/bcbio/bcbio-nextgen) and the VarDict variant caller[33], using as input PCR-deduplicated BAM files and the genomic target regions of the Agilent SureSelect Human All Exon V6 kit. Default parameters were used in the processing pipeline. To summarize the results of the analysis, we used MultiQC[34] and SnpEff (http://snpeff.sourceforge.net/).

**In silico analysis of different restriction enzymes**. To calculate the number of recognition sites in the human reference genome (GRCh37/ hg19) for a variety of restriction enzymes, we used the Restriction module2 package in Biopython[35]. We then used a custom R script to compute the distribution of all the distances between consecutive recognition sites for all the restriction enzymes considered.

**Overlap between HindIII and NlaIII sites and COSMIC genes**. To calculate the number of HindIII and NlaIII recognition sites inside the exons of cancer-associated genes listed in the COSMIC database[15], we first extracted the sequences of all the exons of these genes from the reference human genome (GRCh37/hg19), using the coordinates of the genomic regions targeted by the Agilent SureSelect Human All Exon V6 kit. As sequencing reads can reach exons from restriction sites located in close proximity, we artificially shifted the exons start and end coordinates of 50 bp outwards. Subsequently, we created a FASTA file from these regions using the *BEDTools*[36] function getfasta1. To count the number of restriction sites present inside the extracted exon sequences, we used the Restriction module2 package in Biopython[35]. Lastly, we computed the number of restriction sites per kilobase of exonic sequence. We then used a custom R script to compute the distribution of all the distances between consecutive recognition sites for all the restriction enzymes considered. To measure whether the distribution of the recognition sites is homogenous across the human genome, we binned the human reference genome in 100 kb bins and calculated the SD of the distances between consecutive recognition sites within each bin.

## Data availability

All the sequencing data related to the cell lines described in this study have been deposited in the NCBI Sequence Read Archive with the following SRA accession: PRJNA513980. The sequencing data related to the clinical samples are not disclosed due to patient privacy protection rules enforced at the institutions that provided the samples. The source data underlying all the Figures and Supplementary Figures are provided as a separate Source Data file. The following publically available datasets were used:
1) Human reference genome (Grch37/hg19):

ENSEMBL release 75: http://grch37.ensembl.org/index.html
2) COSMIC gene set: https://cancer.sanger.ac.uk/census
3) List of genes frequently affected by copy number alterations in breast cancers: Supplementary Table 14, sheet "CopyNumber" in ref. [16].
4) List of mutations in 127 genes recurrently mutated in breast cancers: Supplementary Table 2 in ref. [18].
All other relevant data are available from the authors upon resonable request.

## Code availability

All the custom code used for processing CUTseq sequencing data is provided as Supplementary Software and is also available at the following GitHub link: https://github.com/garner1/cutseq.

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

## Acknowledgements

We thank Bastiaan Spanjaard (MDC Berlin) and Alexander van Oudenaarden for support during the early phase of CUTseq development, and Tomasz Kallas (Bienko-Crosetto lab) for help with sequencing. This work was supported by a postdoctoral scholarship from the Swedish Society for Medical Research (SSMF) to X.Z.; by a Mobilność Plus scholarship from the Polish Ministry of Science and High Education (1622/MOB/V/2017/0), and START scholarship from the Foundation for Polish Science (FNP) to M.N.; by grants from Italian Ministry of Health, Ricerca Corrente 2018-2019, AIRC 5xMille MCO Extension Program (# 9970), FPRC 5xMille Ministero Salute 2013, 2014 and 2015 to A.S.; by funding from "Dipartimenti di Eccellenza 2018-2022"(D15D18000410001) to the Department of Medical Sciences of the University of Turin; by additional grants from SSMF, the Swedish Cancer Fund, Stockholm Cancer Fund, and Stockholm County Council to J.H.; by grants from the Science for Life Laboratory, the Swedish Research Council (621-2014-5503), and the Ragnar Söderberg Foundation to M.B.; and by grants from the Swedish Research Council (521-2014-2866), the Swedish Cancer Research Foundation (CAN 2015/585), the Ragnar Söderberg Foundation, the Swedish Foundation for Strategic Research (BD15-0095), and the Strategic Research Programme in Cancer (StratCan) at Karolinska Institutet to N.C. Open access funding provided by Karolinska Institute

## Author contributions

Conceptualization: X.Z., M.B., and N.C. Clinical samples: C.M., J.H., T.V., and A.S. Data curation: X.Z., S.G., and M.S. Formal analysis: S.G. and L.H. Funding acquisition: N.C. and M.B. Investigation: X.Z., M.S., M.N., C.M., and T.V. Methodology: X.Z., N.C., and M.B. Project administration: N.C. Software: S.G. Supervision: N.C. and M.B. Validation: X.Z., M.N., M.S., and R.M. Visualization: S.G., L.H., X.Z., M.S., N.C., and M.B. Writing: N.C., X.Z., and M.B.

## Competing interests

J.H. is a scientific advisor of Visiopharm AG and has been in advisory boards for AstraZeneca and Novartis. He has obtained speaker honoraria from Roche/FoundationMedicine. All the other authors declare no competing interests.
