## [Peer Review File · Nature Communications]

Reviewers' Comments:

Reviewer #1:

Remarks to the Author:

The authors present CUT-seq, a new protocol for preparing NGS libraries. The approach takes a distinct amplification strategy, employing a T7 phage promoter-driven amplification instead of standard PCR amplification, that is the key part. The authors applied CUT-seq to DNA extracted from FFPE sections and compared with standard library preparation, with some cell lines to validate. Cut-seq is entirely a wet laboratory implementation (note not drylab/bioinformatics)

Specifically, CUT-seq first fragments the genome with the restriction enzymes NlaIII or HindIII, followed by ligation to the cut sites together with a standard barcode sequence and the T7 phage minimal promoter sequence. The fragmented DNAs are then amplified by T7 promoter-driven in vitro transcription (ITV). Experiments indicate that CUT-seq is capable of obtaining reasonable quality data from a small amount of DNA (<10 mm²) extracted from FFPE. What? how much DNA are we talking here, DNA yield is VERY cell density, hence tumor (sub-)type dependent!

Though the alternative amplification strategy is interesting, the current manuscript lacks a head to head comparison between standard NGS technology and CUT-seq lab technology. It is not entirely clear what isn't possible with the current PCR-based approaches, and to what extent the CUT-seq adds to the current repertoire. They do not have any support leverage, for ie pcr artefacts such as duplicates whereas they should have this information in hand The manuscript is in general well written, but it should be improved to present better the key improvement over current lab technology of the method. We haven't been able to judge if and how big the improvement is. At the same time there is nothing "wrong" and see a genuine manuscript. I'd like the authors to realize that there is PCR/amplification free NGS procedures out there and in clinical use, that work for FFPE materials (something to take along in comparisons, at least the discussion)

Major comments:

1. CUT-seq removes duplicate reads based on barcodes. It would be interesting to check if T7 phage based-amplification introduced bias based on the filtered duplicates. It can also be compared with data with PCR-amplified sequence data.
2. The authors claimed T7 promoter-driven in vitro transcription (ITV) is linear without any support.
3. The authors showed CUT-seq could work with a small number of DNAs. However, it is also possible to perform the standard workflow with a large number of PCR cycles, which likely to work with small DNAs. Is there a known fact the on the maximum cycle of PCR amplification without too much bias? The authors at least calculate theoretical amount of DNAs to work with standard library preparation with maximum PCR cycles. Also, it would be interesting to check if increased PCR cycles can introduce bias in copy number alteration profiles (and if that is not the case with CUT-seq).
4. In the introduction, authors mention that it is important to develop methods that combine imaging and DNA sequencing of the same tissue sample. However, it is not clear if this is not possible with the standard workflow of sequencing. If this study is the first to obtain both imaging and DNA sequencing from the same sample, the authors should mention that more clearly.
5. It is necessary to indicate why the authors selected either NlaIII or HindIII as a restriction enzyme for each experiment. Currently, it sounds arbitrary.

Minor comments:

1. Numbering in the last paragraph of introduction should be fixed. (two times 2).
2. Figure S4 – "KI-15" is misplaced
3. Figure S5C – Need to explain the color code of the bars.
4. In Figure 3b, useful to highlight the regions mentioned in the text (e.g., chr17q22-q24)

Reviewer #2:

Remarks to the Author:

The authors describe a novel method, CUTseq, that can be used to perform reduced representation sequencing of DNA from tissue sections or sub-region(s). The method development and data presented focus on FFPE material, which is of particular relevance to Pathology laboratories but the authors also note that the method can be applied to fresh-frozen material.

Following isolation from stained histological sections, DNA is cut with one of a set of CUTseq restriction endonucleases and subsequently ligated to barcoded adapters containing a sample-specific barcode (SB), a unique molecular index (UMI), an Illumina adapter and a T7 promoter. T7 RNA polymerase is then used to transcribe and therefore amplify CUT-seq molecules, which are then converted to a sequencing library. Importantly, the SB allows for sample multiplexing prior to T7-driven amplification such that many samples can be processed in the same downstream reactions for improved throughput. The UMI allows for effective filtering of PCR duplicates.

The method was developed using cell line DNA and further validated using 17 FFPE samples in which a subset were processed either by CUTseq or conventional DNA library construction. Importantly, CUTseq was shown to be reproducible and also capable of generating copy number profiles similar to those generated by existing methods. Finally, the authors combine histology with CUTseq to demonstrate 1) the utility of the SB to combine multiple samples (either whole section or sub-regions) early in the workflow; 2) the reproducibility of CUTseq; 3) the robustness of CUTseq in working with archival FFPE material (7-26 years) and 4) the ability of CUTseq (when combined with up front histology) to identify spatial patterns of copy number variation.

The authors provide a list of suitable endonucleases together with corresponding adapter sequence designs and a detailed protocol.

The manuscript is sound, generally well written, readable and accompanied by clear Figures. There is enough detail provided to allow other users to implement CUTseq in their own lab. However, it may be worth considering an additional 'Nature Protocols' publication that brings the supplementary material to the fore, with additional commentary.

I have a number of minor comments:

1. Throughout the paper (Abstract, lines 1-4; Intro, line 11-13; and Discussion, lines 2-4), the authors assert that combining current NGS methods with tissue sections or regions thereof, that have undergone histology is unachievable. In the Introduction, the authors even state that it is not possible with existing methods to generate seq libraries using DNA extracted from FFPE sections that have undergone conventional histology. My concern is that the message here is misleading as there is a repeated suggestion that DNA isolated from tissue that has been stained and imaged is not suitable for current NGS applications, which is not true. To give an example, we routinely generate NGS data (human whole genome) from portions of tissue sections that have undergone e.g. H&E staining followed by laser capture, albeit with fresh-frozen sections. Yes, it is challenging but only because the amount of DNA recovered is low, not because the DNA has come from a tissue that has undergone staining and imaging. The authors should absolutely promote that CUTseq readily enables combining imaging and NGS to gain spatial information, but they should keep it positive and dial back on the negative messaging that somehow DNA from samples that have been imaged is not suitable for current NGS methods. It's the amount of DNA recovered that's problematic, not the fact that it's isolated from a tissue that has undergone histology.

2. The CUTseq method is neat and the authors are right to point out the advantage of barcoding DNA early on in the process such that multiple samples can be combined for downstream processing. It greatly streamlines CUTseq. Although I think the authors have used terms such as

'streamlined' and 'high-throughput' judiciously, I do feel it gives an overall impression that the protocol from imaging to sequencing is readily scalable. One only has to look at the protocol to see that the end-to-end workflow incorporates long-winded, low-throughput, high-labour approaches such as traditional DNA isolation using phenol:ChCl₃ and EtoH precipitation. At some point in the manuscript, the authors should at least highlight such bottlenecks in the process as a limitation. It needn't be presented negatively. For example, it could be presented as an area for further development that has the potential to lead to a more readily scalable process and therefore wider audience.

3. Page 3 line 3. Should read PCR cycles rather than PCR steps.

4. Page 4. Para 2 line 4. Selected regions can be assessed. The limits of the region should be defined here. How small can you go? The authors should provide some guidelines as to what will work with CUTseq and at what point the method will fail.

5. Page 9, line 3. Sub nanogram amounts. Can the authors be specific. Is there a lower limit to input, what approx. input ranges did they use here? Relates to comment 4.

6. Page 5, line 4. There is a large swing in mappability (70.9% +/- 11%) with samples that I would consider unchallenging (cell lines). The authors should comment on this and if it doesn't matter then they should say why.

7. Page 10 line 7. CUTseq can be used to "rapidly" generate highly multiplexed libraries. What is the hands on and total cumulative time for library prep? Relates to comment 2.

8. Could the authors provide costs per sample or at least compare costs to current methods?

9. Page 10, line 9. Please could the authors clarify the role of UMIs? Earlier in the manuscript, UMIs are promoted as allowing removal of PCR duplicate data. Here, at Page 10, line 9, UMIs are said to render the data immune from PCR bias. What do they mean? PCR and bias go hand-in-hand. UMIs allow identification of duplicates and accurate counting of DNA fragments but they don't solve the problem of PCR bias. Please clarify what is meant at this point in the manuscript.

10. Page 11, line 5. The authors refer to a 'Protocol'. I'm not sure I understand whether this is a normal term for something I'm not aware of or whether it was overlooked and meant to be replaced by some more information during manuscript preparation. Either way, please clarify.

Point-by-point response to the Reviewers' comments

Reviewer #1

The authors present CUT-seq, a new protocol for preparing NGS libraries. The approach takes a distinct amplification strategy, employing a T7 phage promoter-driven amplification instead of standard PCR amplification, that is the key part. The authors applied CUT-seq to DNA extracted from FFPE sections and compared with standard library preparation, with some cell lines to validate. Cut-seq is entirely a wet laboratory implementation (note not drylab/bioinformatics)

Specifically, CUT-seq first fragments the genome with the restriction enzymes NlaIII or HindIII, followed by ligation to the cut sites together with a standard barcode sequence and the T7 phage minimal promoter sequence. The fragmented DNAs are then amplified by T7 promoter-driven in vitro transcription (ITV). Experiments indicate that CUT-seq is capable of obtaining reasonable quality data from a small amount of DNA (<10 mm²) extracted from FFPE. What? how much DNA are we talking here, DNA yield is VERY cell density, hence tumor (sub-)type dependent!

Though the alternative amplification strategy is interesting, the current manuscript lacks a head to head comparison between standard NGS technology and CUT-seq lab technology. It is not entirely clear what isn't possible with the current PCR-based approaches, and to what extent the CUT-seq adds to the current repertoire. They do not have any support leverage, for ie pcr artefacts such as duplicates whereas they should have this information in hand The manuscript is in general well written, but it should be improved to present better the key improvement over current lab technology of the method. We haven't been able to judge if and how big the improvement is. At the same time there is nothing "wrong" and see a genuine manuscript. I'd like the authors to realize that there is PCR/amplification free NGS procedures out there and in clinical use, that work for FFPE materials (something to take along in comparisons, at least the discussion)

We are grateful to the Reviewer for appreciating our work and for providing constructive comments and suggestions, which have helped us to considerably strengthen our original manuscript. Following the Reviewer's suggestion, we have now performed a much more thorough assessment of the technical performance of CUTseq, and benchmarked it against widely used commercial kits for NGS library preparation. Specifically:

- 1) We have assessed the reproducibility of DNA copy number profiles for various input amounts of the same gDNA, and number of PCR cycles. As we show in the **new Fig. 1e and Suppl. Fig. 3b-e**, for inputs ranging from 30 ng down to 6 pg of gDNA extracted from the same sample, the quality of sequencing data and the correlations between DNA copy number profiles obtained by CUTseq were highly stable, highlighting the reproducibility of our method.
- 2) Furthermore, as shown in the **new Fig. 2a and 2d**, side-by-side comparison of CUTseq and the commercial kit NEBNext, using the same gDNA series, the corresponding genome-wide DNA copy number profiles were highly correlated. Notably, while with NEBNext we did not succeed, despite several attempts, to build libraries from less than 1 ng gDNA — which is the minimum amount of gDNA recommended by the manufacturer — in contrast, with CUTseq, we succeeded in sequencing samples of 0.5, 0.25, 0.12 and 0.06 ng, by pooling them together into a single, multiplexed library. Notably, the DNA copy number profile of the 0.06 ng sample (corresponding to approx. 10 diploid cell genomes) is still relatively strongly correlated with the profile obtained using 500 times more gDNA input (Pearson's $r =$

0.7, see **new Fig. 1e**), clearly demonstrating the superiority of CUTseq over the standard approach, in making libraries from sub-nanogram samples.

- 3) To further benchmark CUTseq, we compared CUTseq libraries with libraries generated using another commercially available kit for library preparation, namely the SureSelect XT HS Reagent Kit from Agilent Technologies. This time we compared the two kits for their utility in the exome capture procedure. As shown in the **new Fig. 2f-g**, CUTseq and Agilent libraries are largely similar in terms of genome-wide distributions of called nucleotide variants, even though we performed only shallow sequencing. Remarkably however, the coverage of target regions was substantially higher in the case of CUTseq compared to Agilent libraries (80% vs. 52%, see **new Fig. 2h**). Related to this, while CUTseq provides a reduced representation of the exome compared to standard methods that use random genome fragmentation, our analyses suggest that using a frequent cutter and paired-end long reads should allow to cover most of the sites in the human exome that are frequently mutated in cancers (see **new Suppl. Fig. 8**). Combining multiple restriction enzymes should further increase the CUTseq coverage, making it fully comparable, and possibly surpassing many standard whole-exome sequencing approaches.
- 4) We investigated the effect of different IVT durations and number of PCR cycles on the quality of sequence reads and reproducibility of DNA copy number profiles. As shown in the **new Fig. 1f-g and Suppl. Fig. 3f-m**, libraries obtained from the same input amount of genomic DNA (gDNA), but performing IVT for 1, 2, 4, or 14 hours, or including 2, 4, 6, or 8 more PCR cycles result in sequencing results of similar high quality, as well as in highly correlated DNA copy number profiles.
- 5) We have investigated the reproducibility of DNA copy number profiles at various sequencing depths and resolutions, by deeply sequencing one CUTseq library newly generated from the SKBR3 breast cancer cell line, and downsampling the aligned reads. As shown in the **new Fig. 1h-i and Suppl. Fig. 4a-c**, DNA copy number profiles remained highly correlated even at low sequencing depths (less than 300K reads) and relatively high resolution (less than 100 kb). Importantly, the resolution achievable using CUTseq allows for reproducible detection of small focal amplifications and deletions, demonstrating the broad applicability of the method to study copy number alterations of different sizes.
- 6) Regarding the Reviewer's comment on the amount of gDNA extracted from small regions in FFPE tumor sections, we fully agree that, in our original manuscript, we did not address this important issue in a quantitative way. In the **new Suppl. Fig. 6a**, we now provide a precise measurement of the size of the regions from which gDNA was extracted in all the archival FFPE samples now shown in the **new Fig. 4**. A certified pathologist systematically reviewed the same regions, to confirm that they mainly contained tumor cells and not stroma. However, due to the nature of the samples (some of the samples were 27 yrs. old), we were not able to apply our recently developed pipeline for automatic nucleus segmentation and cell counting in FFPE samples (unpublished). To overcome this bottleneck, we instead applied this pipeline to count cells within pathologist-confirmed tumor-rich regions of the same size as the regions from which we extracted gDNA, in a separate cohort of 16 FFPE breast cancer samples. The results, which are presented in the **new Suppl. Fig. 6b-c**, suggest that these regions typically contain between 5,000 and 25,000 cells. We note that, based on the new benchmarking experiment which we described above (see **new Fig. 1e**), we now prove that CUTseq is able to reliably measure the genome-wide DNA copy profile from as little as 60 picograms of gDNA, which corresponds to approx. 10 diploid cells. Therefore, even in regions containing stromal areas and therefore only a few hundreds of cells, CUTseq should be able to reliably profile copy

number levels, as indeed we demonstrate in our multi-region sequencing experiment shown in the **new Fig. 4**.

- 7) Lastly, concerning the possibility to use PCR-free methods to sequence DNA extracted from small regions in FFPE tissue sections, we have performed a systematic review of the commercial options available. As shown in the following table, most kits require a high amount of input gDNA, which cannot be retrieved from single FFPE sections or even smaller regions thereof, as we do in our manuscript. Importantly, the manual of some of these kits (for example, Illumina's TruSeq DNA PCR-free kit) specifically states that the kit is not compatible with FFPE samples.

Kit	Brand	gDNA input (ng)
Illumina TruSeq DNA PCR-Free	Illumina (cat. no. 20015962)	1000-2000
NEXTFLEX PCR-Free DNA-seq kit	PerkinElmer (cat. no. NOVA-5142-01)	500-3000
PCR-free NGS DNA Library Prep kit	BioDynamis (cat. no. 30040S)	100-1000
NxSeq® AmpFREE Low DNA library kit	Lucigen (cat. no. 14000-1)	75
KAPA HyperPlus Library Preparation kit PCR-free	Roche (cat. no. 07961871001)	100-1000
Accel-NGS 2S PCR-free DNA library kit	Swift Biosciences (cat. no. 20024)	100

Instead, commercially available kits compatible with FFPE samples can work with much less gDNA input, but entail the use of PCR to amplify the initial material and to index the library, as shown in the following table, which we now include as **new Supplementary Table 4**. In general, the number of PCR cycles required in these methods depends on the initial gDNA input. For samples in which there is a lot of available DNA, a few PCR cycles are sufficient, and, to this respect, CUTseq is not different from standard methods. However, in the case of individual FFPE tissue sections or smaller regions thereof, the initial amount of gDNA is too low to keep the number of PCR cycles at the minimum. Still, we believe that it is the initial amplification of DNA by in vitro transcription, that allows us to reliably amplify DNA from a minute amounts of input material, not amenable to other procedures.

Kit	Brand	gDNA input (ng)	gDNA input → # PCR cycles
QIAseq Ultralow Input Library	QIAGEN (cat. no. 180492)	0.010-100	10 pg → 16 100 pg → 14 1 ng → 10 10 ng → 8
TruSeq Nano DNA Library Prep kit	Illumina (cat. no. 20015964)	100	100 ng → 8
SMARTer ThruPLEX DNA-seq kit	TaKaRa (cat. no. R400675)	0.050-50	50 pg → 15-16 200 pg → 14-15 1 ng → 11-12 2 ng → 8-10 5 ng → 7-9 10 ng → 7-8 20 ng → 7-8 50 ng → 6-8

NEBNext Ultra™ DNA Library Prep kit for Illumina	NEB (cat. no. E7370S)	0.5-1,000	5 ng → 12 50 ng → 7-8 1 µg → 4
KAPA Hyper Prep kit	Roche (cat. no. KK8503)	1-1,000	500 pg → 12-13 1 ng → 11-12 5 ng → 8-10 10 ng → 7-8 50 ng → 5-6 100 ng → 3-4 500 ng → 1-2
NxSeq UltraLow DNA Library kit	Lucigen (cat. no. 15012-1)	0.050-75	50–250 pg → 15-16 251–750 pg → 13-14 751pg-10 ng → 8-12 11 ng-75 ng → 5-7
Topomize DNA Library Prep kits	MCLAB (cat. no. TOPO-100A)	10-1,000	10 ng → 10-12 50 ng → 8-10 100 ng → 6-8 250 ng → 4-6 500 ng → 3-5 1 µg → 2-4
SureSelect XT HS Reagent Kit with indexes	Agilent (cat. no. G9702A)	10-200	Fresh tissue 10 ng → 11 50 ng → 9 100-200 ng → 8 FFPE tissue 10 ng → 14 50 ng → 12 100-200 ng → 11

We have now empirically assessed the number of PCR cycles required in CUTseq to obtain usable libraries (i.e., libraries in quantities that are compatible with the requirements of Illumina's NextSeq 500 system) for different gDNA inputs (assuming the duration of IVT to be 14 hours). As discussed above, our new results shown in the **new Fig. 1e-g**, demonstrate that, for DNA inputs in the nanogram range, the quality of sequencing reads and the correlations between DNA copy number profiles do not significantly vary by performing between 10 and 18 PCR cycles. When higher amounts of input gDNA are available, the number of cycles can be further decreased: for example, we use 6 cycles for inputs in the range of 500–700 ng, as shown in the **new Suppl. Table 3** included below. We also provide the same table in the step-by-step protocol included in the **Suppl. Methods** and uploaded at **Protocol Exchange** (doi.org/10.21203/rs.2.1742/v1).

Input gDNA (ng)	# PCR cycles
500–700	6
300–400	7
100–200	8
50–100	9
30	10
15	11
7.5	12
3.8	13
1.9	14

Major comments:

1. CUT-seq removes duplicate reads based on barcodes. It would be interesting to check if T7 phage based-amplification introduced bias based on the filtered duplicates. It can also be compared with data with PCR-amplified sequence data.

We thank the Reviewer for raising this important issue. The T7-based method, which we have developed for DNA, builds on method named CELseq, that has been previously developed and thoroughly validated for poly(A) mRNA sequencing in single cells (Hashimshony et al, Cell Rep 2012 and Hashimshony et al, Genome Biol 2016). In both these papers, CELseq was shown to outperform the STRT method for PCR-based multiplexed single-cell RNA-seq (Islam et al, Genome Res 2011), both in terms of number of genes per cell as well as in terms of noise across biological replicates. As already discussed above, we have now shown that the IVT duration and number of PCR cycles have very little effects (if any) on the quality of sequencing data and reproducibility of DNA copy number profiles obtained by CUTseq (see **new Fig. 1e-g**). Furthermore, as also discussed above, the results of the new exome capture experiment shown in **Fig. 2f-h** additionally confirms that the presence of the IVT step in the CUTseq protocol does not negatively affect the target region coverage and genome-wide distribution of single-nucleotide variants.

2. The authors claimed T7 promoter-driven in vitro transcription (ITV) is linear without any support.

The linearity of in vitro transcription by the T7 phage RNA polymerase has been shown before in multiple publications, including the very first paper documenting the use of IVT for whole-mRNA amplification from single cells (Eberwine et al., PNAS 1992), as well as the original paper describing the CELseq method mentioned above (Hashimshony et al, Cell Rep 2012). However, we agree with the referee that our experimental conditions might change the reaction kinetics and therefore it is not necessarily fair to extrapolate previous observations to our case. For this reason, and since our main goal is to demonstrate that CUTseq is a robust and efficient method for the preparation of multiplexed DNA sequencing libraries, rather than characterizing in depth the properties of the IVT reaction, in the revised manuscript we now refrain from stating that the IVT amplification reaction is linear. Instead, we highlight the fact that, thanks to IVT, even picogram quantities of gDNA can be efficiently barcoded in a two-step reaction (digestion and ligation) that does not require intermediate DNA purification, which would cause unavoidable loss of DNA (see **new Fig. 1e and 2a**). This could not be achieved using a PCR-only based method, because each DNA fragment would have to be simultaneously ligated to two different adapters, one on each fragment extremity, which is probabilistically less favorable than a single ligation event. This means that, especially in low-input samples, the majority of DNA fragments would only be ligated to one adapter, and therefore would not be amplified during the subsequent PCR steps. In contrast, using IVT, any DNA fragment, that has been successfully ligated to a linker containing the T7 promoter sequence and the RA5 sequencing adapter, has a chance to be amplified in sufficient quantity so that also the second RA3 adapter can be attached to it. As we now point out in the revised Discussion, this fundamental difference also distinguishes CUTseq from other reduced representation sequencing methods, such as RAD-seq (Baird et al, PLoS One 2008), that use PCR to introduce both sequencing adapters, and therefore require a higher gDNA input.

3. The authors showed CUT-seq could work with a small number of DNAs. However, it is also possible to perform the standard workflow with a large number of PCR cycles, which likely to work with small DNAs. Is there a known fact the on the maximum cycle of PCR

amplification without too much bias? The authors at least calculate theoretical amount of DNAs to work with standard library preparation with maximum PCR cycles. Also, it would be interesting to check if increased PCR cycles can introduce bias in copy number alteration profiles (and if that is not the case with CUT-seq).

As we have amply discussed above, we have now performed a series of experiments in which we compared different number of PCR cycles, and provide an empirically based recommendation for the number of PCR cycles to use for different amounts of input gDNA (see **new Suppl. Table 3**). Although, in principle, the Referee is right in suggesting that, perhaps, the same results could be obtained using a standard library preparation kit, simply by increasing the number of PCR cycles, we note that we have been unable to generate detectable libraries from gDNA inputs smaller than 1 ng using a widely used standard kit (NEBNext), despite multiple attempts, even using more than 16 PCR cycles. The likely explanation of the lack of success goes back to our previous response: PCR-based only methods need to attach adapters on both ends of DNA fragments. This is, per se, probabilistically less favorable than attaching only one adapter on one end, as in CUTseq. Furthermore, standard PCR-based methods entail DNA fragmentation by sonication, which requires end-repair and an additional DNA purification step before adapter ligation. In contrast, in CUTseq, the input gDNA is digested and ligated in the same tube, without any intermediate purification step. We note that, in principle, it is also possible to perform the IVT reaction in the same tube, simply by topping up the reaction volume, although we prefer to pool multiple samples together and purify them before making a single IVT reaction.

4. In the introduction, authors mention that it is important to develop methods that combine imaging and DNA sequencing of the same tissue sample. However, it is not clear if this is not possible with the standard workflow of sequencing. If this study is the first to obtain both imaging and DNA sequencing from the same sample, the authors should mention that more clearly.

This is an important point, also raised by Reviewer #2. Indeed, we were also able to prepare a library from a single FFPE tumor section after imaging it, using the standard NEBNext kit. Therefore, both Reviewers are right in pointing out that the integration of imaging and sequencing is not a unique feature of CUTseq. Accordingly, we have rephrased our original manuscript, aiming to highlight what we believe represents the true innovative aspect of CUTseq, i.e. its ability to barcode DNA at an early stage during library preparation, and therefore generate highly multiplexed libraries in a streamlined and cost-efficient manner from low-input material. In further support of this, we have now implemented a semi-automated workflow that allows preparing multiplexed CUTseq libraries containing up to 96 samples of few nanograms, in only 8 hours (new **Fig. 3a-b**). As shown in the **new Fig. 3c-d and Suppl. Fig. 5**, the copy number profiles of 96 technical replicates prepared in such manner are highly reproducible, while the sequencing error rate is uniformly low (**new Fig. 3e**). Importantly, the preparation of highly multiplexed libraries using this high-throughput CUTseq platform is much more cost-effective compared to standard NGS library preparation technologies, especially when dealing with large numbers of samples (see the cost analysis presented in the **new Suppl. Note 1**). As we point out in the Discussion, high-throughput CUTseq will be particularly useful in the context of high-throughput CRISPR screens and cell line authentication efforts, as a simple and cost-effective strategy to monitor for DNA copy number changes in a large number of samples in parallel.

5. It is necessary to indicate why the authors selected either NlaIII or HindIII as a restriction enzyme for each experiment. Currently, it sounds arbitrary.

We have now clarified this aspect, by adding the following part in the Results section:

“We selected NlaIII and HindIII based on the frequency and distribution of their recognition sites along the genome, and given their lower cost compared to other enzymes in the same class (**Supplementary Fig. 1a-d and Supplementary Table 1**).”

The choice of which of the two enzymes to use in different experiments throughout the manuscript was mainly dictated by practical considerations. For the multi-region tumor sequencing experiment, as well as for the high-throughput CUTseq experiment shown in the **new Fig. 3**, we purchased 96 differently barcoded CUTseq adapters, and chose HindIII in order to achieve a higher depth per recognition site, given the fact that this enzyme cuts less frequently compared to NlaIII (see **new Suppl. Fig. 1**). For copy number profiling, HindIII and NlaIII are largely interchangeable, although the cost per reaction is higher in the case of NlaIII (see **new Suppl. Table 5**). In contrast, for exome profiling, NlaIII is preferable to HindIII, as it provides a much higher coverage, as shown in the **new Suppl. Fig. 1 and 8**.

Minor comments:

1. Numbering in the last paragraph of introduction should be fixed. (two times 2).

This part has now been changed in the revised manuscript, and moved to the Discussion section.

2. Figure S4 – “KI-15” is misplaced

This figure has now been modified and moved to a **new Fig. 4** in the revised manuscript.

3. Figure S5C – Need to explain the color code of the bars.

This figure has now been modified and incorporated into the new **Fig. 4b-c**. We now show that the aneuploid genome fraction (i.e., percentage of the genome either amplified or deleted) is very highly correlated (Pearson's $r = 0.98$) between L1 and L2 technical replicate CUTseq libraries derived from DNA extracted from $\sim 7 \text{ mm}^2$ regions inside individual tissue sections of primary and metastatic breast cancer lesions.

4. In Figure 3b, useful to highlight the regions mentioned in the text (e.g., chr17q22-q24)

We have performed a much more in-depth analysis of the DNA copy number profiles obtained from different regions in primary and metastatic breast cancer lesions, and assessed the prevalence of amplifications and deletions affecting 712 genes listed in the COSMIC database (<https://cancer.sanger.ac.uk/cosmic>), which are frequently mutated across many cancer types. As shown in the **new Fig. 4g-h**, we found that the samples that we examined formed two distinct groups, one characterized by both amplifications and deletions affecting many COSMIC genes, and the other mainly characterized by amplifications. In the latter group, MYC was the most frequently amplified gene, and many samples also contained an amplification of the ERBB2/HER2 gene. In our revised manuscript, we now discuss in depth these and other findings, which we believe convincingly demonstrate the utility of CUTseq for studying intratumor genetic heterogeneity, at a spatial resolution that has so far not been possible to achieve with classical multi-region tumor sequencing approaches.

Reviewer #2

The authors describe a novel method, CUTseq, that can be used to perform reduced representation sequencing of DNA from tissue sections or sub-region(s). The method development and data presented focus on FFPE material, which is of particular relevance to Pathology laboratories but the authors also note that the method can be applied to fresh-frozen material.

Following isolation from stained histological sections, DNA is cut with one of a set of CUTseq restriction endonucleases and subsequently ligated to barcoded adapters containing a sample-specific barcode (SB), a unique molecular index (UMI), an Illumina adapter and a T7 promoter. T7 RNA polymerase is then used to transcribe and therefore amplify CUT-seq molecules, which are then converted to a sequencing library. Importantly, the SB allows for sample multiplexing prior to T7-driven amplification such that many samples can be processed in the same downstream reactions for improved throughput. The UMI allows for effective filtering of PCR duplicates.

The method was developed using cell line DNA and further validated using 17 FFPE samples in which a subset were processed either by CUTseq or conventional DNA library construction. Importantly, CUTseq was shown to be reproducible and also capable of generating copy number profiles similar to those generated by existing methods. Finally, the authors combine histology with CUTseq to demonstrate 1) the utility of the SB to combine multiple samples (either whole section or sub-regions) early in the workflow; 2) the reproducibility of CUTseq; 3) the robustness of CUTseq in working with archival FFPE material (7-26 years) and 4) the ability of CUTseq (when combined with up front histology) to identify spatial patterns of copy number variation.

The authors provide a list of suitable endonucleases together with corresponding adapter sequence designs and a detailed protocol.

The manuscript is sound, generally well written, readable and accompanied by clear Figures. There is enough detail provided to allow other users to implement CUTseq in their own lab. However, it may be worth considering an additional 'Nature Protocols' publication that brings the supplementary material to the fore, with additional commentary.

We are very grateful to the Reviewer for appreciating the quality of our work and for recognizing the advantages of CUTseq over existing methods for DNA sequencing library preparation. Following the Referee's suggestions, we have now uploaded a step-by-step CUTseq protocol to the [Protocol Exchange](https://doi.org/10.21203/rs.2.1742/v1) website (doi.org/10.21203/rs.2.1742/v1). We have also done a number of significant additions and improvements to the CUTseq protocol described in the previous manuscript, which are now included in the new step-by-step protocol uploaded to Protocol Exchange. Specifically:

- 1) We describe a high-throughput CUTseq workflow, which can be used to prepare highly multiplexed libraries in a short turnaround time and cost-effective manner. We have implemented this workflow using a new contactless nanodispensing system (I-DOT One from Dispensix), which we show in the **new Fig. 3b**. Using this system, we have been able to generate ready-to-sequence CUTseq libraries from 96 DNA samples in 8 hours (see **new Fig. 3a**). In the **new Fig. 3c-e and Supp. Fig. 5**, we now show that the sequencing read quality and DNA copy number profiles of 96 replicates of HeLa cell DNA, obtained using high-throughput CUTseq, are highly similar across replicates, demonstrating the robustness of this approach. Importantly, as we demonstrate in the **new Supplementary Note 1**, high-throughput CUTseq allows preparing libraries from thousands of samples, at a cost that is significantly lower compared to commercially available kits for DNA library preparation. This will provide a major advantage for NGS

facilities than process a large number of samples, as well as for public cell repositories, such as ATCC, as a tool for rapid and cost-effective screening of DNA copy number alterations and genome instability in widely used cell lines.

- 2) Following Reviewer's #1 request, in the revised step-by-step protocol as well as in the **new Suppl. Table 3**, we provide information on how to scale the number of PCR cycles depending on the amount of gDNA input.
- 3) We clearly state that the CUTseq protocol is compatible with any gDNA purification strategy, and provide a number of different options which we have tested (see also response below to the question on phenol-chloroform).

I have a number of minor comments:

1. Throughout the paper (Abstract, lines 1-4; Intro, line 11-13; and Discussion, lines 2-4), the authors assert that combining current NGS methods with tissue sections or regions thereof, that have undergone histology is unachievable. In the Introduction, the authors even state that it is not possible with existing methods to generate seq libraries using DNA extracted from FFPE sections that have undergone conventional histology. My concern is that the message here is misleading as there is a repeated suggestion that DNA isolated from tissue that has been stained and imaged is not suitable for current NGS applications, which is not true. To give an example, we routinely generate NGS data (human whole genome) from portions of tissue sections that have undergone e.g. H&E staining followed by laser capture, albeit with fresh-frozen sections. Yes, it is challenging but only because the amount of DNA recovered is low, not because the DNA has come from a tissue that has undergone staining and imaging. The authors should absolutely promote that CUTseq readily enables combining imaging and NGS to gain spatial information, but they should keep it positive and dial back on the negative messaging that somehow DNA from samples that have been imaged is not suitable for current NGS methods. It's the amount of DNA recovered that's problematic, not the fact that it's isolated from a tissue that has undergone histology.

We completely agree with the Reviewer that the true challenge of working with small tissue samples is not the fact that they have been stained prior to DNA extraction, but rather that the amount of gDNA that can be recovered is often too little for a successful preparation of NGS libraries using standard methods. In this context, the true novelty of CUTseq, as the Reviewer rightly points out, lies in the ability to barcode DNA very early during library preparation, which in turns allows for multiplexing and thus substantially reducing costs, as we now show in the **new Suppl. Note 1**. Importantly, multiplexing is also critical for sequencing of low-input DNA samples (due to the crowding effect of pooling of DNA from multiple samples) for which it would not be possible to generate libraries using the existing technology. Indeed, as we show in the **new Fig. 1e**, this key feature of CUTseq allowed us to obtain a reliable DNA copy number profile from as little as 60 pg of gDNA extracted from a single FFPE breast cancer section, from which we did not succeed to prepare a library using the commercially available and widely used NEBNext kit. Accordingly, we have now changed the title and the manuscript throughout, in order to highlight this unique aspect of CUTseq, while we still show the full compatibility of CUTseq with tissue sections that have been previously stained. Related to this aspect, in response to the request of Reviewer's #1, we have additionally applied a fully automated pipeline, which we have recently developed (unpublished), to accurately count the number of cells in the tissue regions equivalent to those sequenced by CUTseq (see **new Suppl. Fig. 6**). In the future, this pipeline will enable to quantitatively relate the sequencing profiles obtained by CUTseq, with the number of cells and tumor purity assessed in the same sample from which gDNA is extracted.

2. The CUTseq method is neat and the authors are right to point out the advantage of barcoding DNA early on in the process such that multiple samples can be combined for downstream processing. It greatly streamlines CUTseq. Although I think the authors have used terms such as 'streamlined' and 'high-throughput' judiciously, I do feel it gives an overall impression that the protocol from imaging to sequencing is readily scalable. One only has to look at the protocol to see that the end-to-end workflow incorporates long-winded, low-throughput, high-labour approaches such as traditional DNA isolation using phenol:ChCl3 and EtoH precipitation. At some point in the manuscript, the authors should at least highlight such bottlenecks in the process as a limitation. It needn't be presented negatively. For example, it could be presented as an area for further development that has the potential to lead to a more readily scalable process and therefore wider audience.

We thank the Reviewer for raising this point. As we have already mentioned above, in the **revised step-by-step CUTseq protocol**, which we now provide in the Suppl. Methods and at Protocol Exchange (doi.org/10.21203/rs.2.1742/v1), we clearly state that the in vitro CUTseq workflow is compatible with gDNA extracted with different procedures, including the more time-consuming phenol-chloroform extraction, as well as faster and simpler silica-based kits. In our experiments, we used different extraction methods, including phenol-chloroform and silica-based columns, which we now describe in the Methods section. The reason why we decided to use old-fashioned phenol-chloroform extraction, despite the procedure being lengthier and more cumbersome, is that, in our experience, this allows retrieving DNA with the highest purity. However, especially for clinical applications, faster silica-based DNA extraction methods are perfectly compatible with CUTseq.

3. Page 3 line 3. Should read PCR cycles rather than PCR steps.

Since we have extensively rephrased the Introduction as well as other parts in the manuscript — to be able to include more results, and to more effectively highlight the novelty and advantages of CUTseq — this sentence has now been removed.

4. Page 4. Para 2 line 4. Selected regions can be assessed. The limits of the region should be defined here. How small can you go? The authors should provide some guidelines as to what will work with CUTseq and at what point the method will fail.

As mentioned above, in the **new Suppl. Fig. 6** we now provide a thorough quantification of the number of cells in regions of the same size such as those in which we have successfully profiled DNA copy number levels, as shown in **Fig. 4**. In 0.4 μm -thick FFPE breast cancer tissue sections, such regions typically contain between 5,000 and 25,000 cells. However, as we show in the **new Fig. 1e**, CUTseq was also able to sequence gDNA derived from 60 pg of gDNA extracted from a single FFPE section, which roughly corresponds to 10 diploid cells. We now describe this aspect thoroughly in the Results and Discussion.

5. Page 9, line 3. Sub nanogram amounts. Can the authors be specific. Is there a lower limit to input, what approx. input ranges did they use here? Relates to comment 4.

As mentioned in the previous response, we now provide a thorough quantification of this in the **new Suppl. Fig. 6**.

6. Page 5, line 4. There is a large swing in mappability (70.9% +/- 11%) with samples that I would consider unchallenging (cell lines). The authors should comment on this and if it doesn't matter then they should say why.

The Reviewer is right in commenting that the mappability varies among libraries made from different cell lines, and indeed this is the case also for libraries made from FFPE samples, as it can be seen in the sequencing statistics for all samples, that we now provide as **new Supplementary Data 2**. In our experience (not only based on CUTseq, but also on several other NGS methods which we have developed [e.g., Crosetto et al, Nature Methods 2013 and Yan et al., Nature Communications 2017]), the mappability can vary quite significantly from one sequencing run to the other, even for the same library, simply because of how the run is performed (we share a NextSeq instrument with other users and often mix different types of libraries in the same run). Importantly, however, a relatively low mappability by no means implies that the sequencing reads with the correct barcode and aligned to the reference genome are of poor quality. In fact, as it can be seen in the **new Suppl. Data 2**, as well as in the **new Fig. 3e** for the aforementioned 96 samples processed with our new high-throughput CUTseq, the sequencing error rate in all our experiments is very homogeneous across replicates, and is typically below 1% in both FFPE and non-fixed samples. Accordingly, we have rephrased the relevant sentence in the paragraph about ‘CUTseq implementation’ as following:

“All the libraries showed a homogeneous fragment size distribution, and yielded a high proportion of reads with the expected prefix ($95.6\% \pm 0.8\%$, mean \pm s.d.), very low error rate ($0.52\% \pm 0.26\%$, mean \pm s.d.), equal partitioning between the Watson and Crick strands, and even distribution of all the four bases at every position along the UMI sequence (**Supplementary Fig. 2a-b and Supplementary Data 2**).”

7. Page 10 line 7. CUTseq can be used to “rapidly” generate highly multiplexed libraries. What is the hands on and total cumulative time for library prep? Relates to comment 2.

As described above, we have now implemented a streamlined workflow, using the I-DOT One contactless nanodispensing system, which allows to obtain a ready-to-sequence library containing 96 different samples in 8 hours. We now highlight this streamlined assay in the main text, and show a schematic timeline in the **new Fig. 3a**.

8. Could the authors provide costs per sample or at least compare costs to current methods?

We have now performed a through cost analysis of the cumulative cost of CUTseq versus eight different commercially available DNA library preparation methods, which are compatible with FFPE samples. Our analysis indicates that, already with a modest multiplexing rate (6 samples per library), CUTseq outperforms all the commercial kits in terms of rate of the cumulative cost curve, as it can be seen in the plots below, which is also included in the **new Suppl. Note 1**.

Legend. The plot on the right shows a magnification of the one on the left, for the first 2,000 samples. The numbers near the ‘CUTseq’ label indicate the numbers of samples pooled into the same library.

We note that, for a very limited number of samples to be processed, CUTseq will have a higher cost compared to commercial kits, because many of the reagents needed for CUTseq can only be purchased in large amounts (e.g., AMPure beads). However, for a

large number of samples to be processed, our multiplexing method has a clear advantage over making single libraries for every sample. We discuss this thoroughly in the **new Suppl. Note 1**, showing how this could be particularly advantageous for NGS facilities as well as cell repositories to screen a large number of cell lines at affordable cost.

9. Page 10, line 9. Please could the authors clarify the role of UMIs? Earlier in the manuscript, UMIs are promoted as allowing removal of PCR duplicate data. Here, at Page 10, line 9, UMIs are said to render the data immune from PCR bias. What do they mean? PCR and bias go hand-in-hand. UMIs allow identification of duplicates and accurate counting of DNA fragments but they don't solve the problem of PCR bias. Please clarify what is meant at this point in the manuscript.

We thank the Reviewer for pointing this out. Indeed, the Reviewer is correct in stating that the role of UMIs is to allow for *in silico* removal of PCR duplicates, without the need for paired-end sequencing. We have now corrected the text accordingly.

10. Page 11, line 5. The authors refer to a 'Protocol'. I'm not sure I understand whether this is a normal term for something I'm not aware of or whether it was overlooked and meant to be replaced by some more information during manuscript preparation. Either way, please clarify.

We apologize for this inaccuracy, we have now corrected this in the revised manuscript.

Reviewers' Comments:

Reviewer #1:

Remarks to the Author:

Overall, the authors significantly improved their manuscript and addressed many of the concerns we raised. Especially, the uniqueness of CUT-seq is presented much more clear by providing a head-to-head comparison with the standard procedures. Nonetheless, there are a few points still unclear with regard to the added value of CUTseq laboratory procedures; We feel they still claim more than is supported by the data. First of all, the authors claim that they can recover genome-wide copy number profiles using as low as 60pg of gDNA input. However, most of the profiles obtained with this low amount of gDNA input (< 100ng cases) have a low number of copy number alterations, it is not unambiguous whether this is due to the low sensitivity of the technology or the low number of aberrations in the samples. Furthermore, it is inconclusive if focal amplifications and SNVs detected by CUT-seq are trust-worthy.

Specific comments

1. Plots for copy number profiles are low in resolution and often difficult to interpret. It is helpful for the readers to add chromosome numbers on the x-axis of the plots and/or plots with a few of the chromosomes to highlight the range of focus, at least for a subset. Especially, in Figure 1e, the sample is generally copy number silent and hence the current plot does not inform readers why correlations get lower as the input gDNA gets smaller. Samples with (more) alterations would be recommended to be able to interpret and compare

2. In line with the above, the authors show a change in performance of the copy number profile estimation with the diverse settings, such as Figure 1e-g, and Figure 2a-b/d. In all these figures, the authors picked a reference to compare with (e.g., gDNA 30ng with 10 PCR cycles in Figure 1e). However, to prove a gradual decrease in performance due to change of setting (e.g. increased PCR cycle), the authors can provide a correlation heatmap, where row and columns are the setting tested (e.g. amount of input gDNA; similar to Supplementary Figure 4b).

3. In Figures 1c-d, there is quite some decrease in performance due to the decreased amount of gDNA from 200ng to 100 ng (for instance, A549). After the cell line experiment, many of the tissue-based experiment used small amount gDNA of 30ng. Also, there are often almost no copy number alterations observed for the samples with a small amount of gDNA (Figure 1e, 1f, 2a). We cannot rule out the possibility that CUT-seq is not sensitive enough with low gDNA input. The authors should try a smaller amount of gDNA for cell lines (in figure 1c) to show the robustness of CUT-seq with small amount of gDNA.

4. The authors claim that CUT-seq can pick up focal amplifications with 10kb resolution, which is questionable (Figure 1h-i). The difference between signal and noise is not convincing. It looks quite noisy when focal amplification is called with 10kb resolution, according to the profiles in Supplementary Figure 4a. The authors can cross-check with CNV profile previously reported for SKBR3 cell lines to prove if the noise-like profile is actually the true focal amplification.

5. The authors compared CUT-seq with exome capture with the standard Agilent exome-seq protocol. The frequencies and genomic distribution of SNVs are comparable between the two technologies (Figure 2f-g), but also between the two samples (TRN15, TRN 18). It is unclear if the technologies are comparable or the measure comparisons (frequency and genomic distribution) are just insensitive to detect the difference. The authors should perform a systematic and higher-resolution comparison to draw a conclusion. (e.g. how many SNVs captured by both technology, and how comparable they are within and between the samples?)

6. The authors explained why they picked the two enzymes, but it is not clear when to use which enzyme. It seems NlaIII is better in terms of coverage and narrower gaps. Is there any benefit of using the other enzyme? Should be discussed in the discussion.

7. Supplementary Figure 4 a-b do not seem to match with each other. The least correlated pairs should appear at the resolution of 10kb according to 4a, but in 4b, lowly correlated pairs appear at two resolutions (10kb and 50kb, with the lowest sequencing depth). And it seems 10kb resolution gives rather unstable copy number profile.

8. Protocol exchange link (<https://doi.org/10.21203/rs.2.1742/v1>) seems to be not active.

Reviewer #2:

Remarks to the Author:

The authors original submission described a novel method, CUTseq, that can be used to perform reduced representation sequencing of DNA from tissue sections or sub-region(s). Isolated DNA is cut with one of a set of CUTseq restriction endonucleases and subsequently ligated to barcoded adapters containing a sample-specific barcode (SB), a unique molecular index (UMI), an Illumina adapter and a T7 promoter. T7 RNA polymerase is then used to transcribe and therefore amplify CUT-seq molecules, which are then converted to a sequencing library by reverse-transcription and PCR. The authors developed the CUTseq method using cell line DNA and further validated it using 17 FFPE samples in which a subset were processed either by CUTseq or conventional DNA library construction.

In my original review of the manuscript, I made a number of comments and I am pleased to say that the authors have addressed each point satisfactorily and together with responses to Reviewer 1, the manuscript is greatly improved. For example, there is 1) greater clarity on the unique benefits of CUTseq; 2) detailed validation data indicating the lower limits (DNA input) of CUTseq and where possible, how it compares with a widely used library method; 3) expected throughput and cost as well as a description/validation of streamlined CUTseq implemented using nano dispensing; 4) a detailed CUTseq protocol resource published on the Protocol Exchange for users wishing to adopt the method.

Comment 1: Supplementary Cost analysis - The authors have validated CUTseq on I-DOT and

based all their cost calculations using this platform. Could the authors clarify that the commercial alternatives are based on recommendations of the manufacturer, please? It's not the authors responsibility to work out or indeed validate various offerings on I-DOT but the reader should be clear that CUTseq costs are benefiting from low volume dispensing whereas the competition is not.

Comment 2: In this revised manuscript, the authors also demonstrate that CUTseq libraries are compatible with exome sequencing, which will be of particular interest to some groups. However, I am unclear whether the authors now include these data to indicate a compatibility (proof-of-principle) of CUTseq with existing exome capture workflows (Page 8, line 25 - Page 9, line 4 and Page 13, lines 13-15) or to promote CUTseq as a genuinely viable alternative to current exome workflows as suggested on P.13, line 11-12 in particular. I appreciate the latter point is about choice of enzyme to improve target coverage but I don't feel that there is enough CUTseq performance data presented to assure the reader that the CUTseq combined with exome Seq could be 'virtually indistinguishable from classical exome sequencing methods', even with the perfect restriction enzyme in hand. I just think this point needs clarifying and in its current form (p. 13, para 1) it's potentially misleading.

Comment 3: The authors show that ~80% of the exome target is covered for 2 DNA samples, TRN17 and TRN18. In contrast, the commercial alternative from Agilent targeted ~50%. The results from the Agilent reagents seem disappointing. Notwithstanding the effect of FFPE, I would have expected better from a 50 ng input. Could the authors provide some comment and/or possible explanation? Were the number of reads equivalent, were there enough reads, how do duplicate rates compare?

Comment 4: Related to Comment 3 - could the authors indicate in the text the fold (depth) coverage across the target? This will be of interest to users wishing to reliably call variants. Although CUTseq is demonstrated to be effective at ~60 pg input for CNA detection, could the authors estimate/predict lowest input for exome Seq?

Please note that Comments 2-4 and their importance/interest will depend upon what message the authors are wishing to convey with respect to CUTseq/exome Seq. I am convinced that CUTseq is compatible with exome capture based on e.g. Fig 2f-h. but I think anyone genuinely interested in implementing CUTseq with exome capture with the expectation of matching existing methods would benefit from additional performance data as indicated in the above comments.

Comment 5: Fig 3c, heat map - 10 samples were omitted for noisy and therefore unreliable profiles. Could the authors offer an explanation?

Comment 6: Supp. Fig. 3d - Duplication rate reaches a plateau (~80%) at ~3ng - any possible explanation?

Point-by-point response to the Reviewers' comments

Reviewer #1

Overall, the authors significantly improved their manuscript and addressed many of the concerns we raised. Especially, the uniqueness of CUT-seq is presented much more clear by providing a head-to-head comparison with the standard procedures. Nonetheless, there are a few points still unclear with regard to the added value of CUTseq laboratory procedures; We feel they still claim more than is supported by the data. First of all, the authors claim that they can recover genome-wide copy number profiles using as low as 60pg of gDNA input. However, most of the profiles obtained with this low amount of gDNA input (< 100ng cases) have a low number of copy number alterations, it is not unambiguous whether this is due to the low sensitivity of the technology or the low number of aberrations in the samples. Furthermore, it is inconclusive if focal amplifications and SNVs detected by CUT-seq are trust-worthy.

We are grateful to the Reviewer for recognizing the improvement of our manuscript as a result of the first revision. We also appreciate the Reviewer's remaining concerns regarding the sensitivity of CUTseq and her/his request for more experiments corroborating the sensitivity and reproducibility of CUTseq, particularly in low-input samples showing copy number alterations (CNAs). To this end, we now have performed several new experiments, and substantially revised both Figure 1 and 2, as well as significantly expanded the Supplementary Figures. The main changes and additions that we have done in this second revision can be summarized as follows:

- We have repeated the experiments with cell lines using a lower gDNA input (30 ng as suggested by the Reviewer) digested with either HindIII and NlaIII. As shown in the **new Fig. 1b-c and Supplementary Fig. 3**, the correlation between corresponding HindIII and NlaIII samples is extremely high for all the cell lines examined, at five different resolutions spanning from 1 Mb down to 30 kb. This shows that we have improved the CUTseq method since its very initial days when we obtained the libraries that we presented in the original manuscript.
- We have included a new breast cancer cell line (BT474), which has a high number of CNAs and is known to carry an amplification of the clinically relevant *ERBB2/HER2* oncogene on chr17. This allowed us to further confirm the specificity of CUTseq, by comparing the *ERBB2* copy number in a total of three breast cancer cell lines with well-established HER2 status: MCF7 cells, which are HER2-negative, and SKBR3 and BT474 cells, which are HER2-positive. As we show in the **new Figure 1d** (in which we visualize copy number profiles along a single chromosome, following this Reviewer's suggestion, see also related comment below), CUTseq specifically and reproducibly detected the *ERBB2* gene as amplified only in BT474 and SKBR3 cells, but not in MCF7 cells, further highlighting the specificity and reproducibility of CUTseq.
- In order to test the sensitivity of CUTseq for sub-nanogram gDNA inputs, we have performed a completely new set of experiments, using gDNA extracted from one FFPE tumor sample that carries many CNAs, and comparing inputs from 1 ng down to 120 pg, and three different PCR cycles (12, 14, and 16). As we show in the **new Fig. 1k, l and Supplementary Fig. 6 and 7**, the correlations between the copy number profiles obtained with different amounts of input DNA and PCR cycles were extremely high, at five different resolutions. Even when using 120 picograms of input, the correlations were above 0.98 (Pearson's ρ) at 30 kb

resolution, highlighting the sensitivity and reproducibility of CUTseq even for picogram amounts of FFPE DNA — which is well below the limit of most commercially available kits. Importantly, extra PCR cycles had no detrimental effect on the profiles, further highlighting the robustness of CUTseq.

- We have repeated the exome capture experiment, by performing two new replicate experiments using gDNA extracted from the highly genomically unstable SKBR3 breast cancer cell line. As we show in the **new Fig. 2c-f and Supplementary Fig. 10c-e**, even though the number of high-confidence SNVs identified by CUTseq was lower in comparison to a standard reference method (Agilent) — which is expected based on the fact that CUTseq is a reduced genome representation method — ~38% of the SNVs were found by both methods. Importantly, ~72% of all the high-confidence SNVs found by CUTseq were identified in both experimental replicates, further confirming the reproducibility of CUTseq. These results clearly show (i) that CUTseq libraries are fully compatible with a standard method for exome capture (Agilent SureSelect), and that (ii) CUTseq provides a reproducible, albeit reduced, exome representation when a single restriction enzyme is used.

As we also discuss below in response to the related Reviewer's comment, it was never our intention to convey the idea that CUTseq would be superior to standard WES methods which rely on random genome fragmentation during library preparation — at least when a single restriction enzyme is used. Moreover, in this manuscript, we only wanted to provide proof-of-principle evidence that CUTseq libraries can be used for reduced exome representation sequencing. As we now write in the revised Discussion, one application of such reduced representation exome sequencing would be in multi-region tumor sequencing, in order to detect less high-confidence SNV events, but from many more regions than currently possible, at comparable sequencing costs. This would significantly improve the ability to reconstruct a tumor's phylogeny, by comparing CNA and SNV profiles from many regions in the same tumor. In the revised manuscript, we now emphasize that this experiment was only a proof-of-concept and that CUTseq can only provide a reduced representation of the exome, also in line with the second Reviewer's comment on this issue.

We would still like to include these results in our final manuscript, as we think they contribute to show the versatility of CUTseq and might be particularly appealing for readers working in the field of tumor heterogeneity. However, if the Reviewer finds that these data do not add much to the manuscript, we would of course remove this part and focus entirely on copy number profiling.

- Following this Reviewer's suggestion (see also related comment below), we have changed all the copy number profile plots (except in Fig. 4 due to space constraints), and now present plots showing raw data as grey dots with overlaid segmented profiles in black, and x-axis ticks marking the boundary between each chromosome. We hope that the Reviewer will find this new graphical representation more compelling and truly reflective of the richness of our dataset and high correlations reported.

In summary, we believe that we have substantially further improved our manuscript, and that we now convincingly show that CUTseq is a sensitive and reproducible method for high-resolution copy number profiling, even in picogram DNA samples from FFPE tissue.

Specific comments

1. Plots for copy number profiles are low in resolution and often difficult to interpret. It is

helpful for the readers to add chromosomes numbers on the x-axis of the plots and/or plots with a few of the chromosomes to highlight the range of focus, at least for a subset. Especially, in Figure 1e, the sample is generally copy number silent and hence the current plot does not inform readers why correlations get lower as the input gDNA gets smaller. Samples with (more) alterations would be recommended to be able to interpret and compare

We thank the Reviewer for this important point and suggestion. As already mentioned above, we have now changed all the copy number plots to show read counts together with segmented profiles, as well as the boundaries between individual chromosomes. Following the Reviewer's suggestion, we now also show individual chromosome profiles in the **new Fig. 1d and 1j**. We have removed the old Fig. 1e and the related dataset, and substituted it with a new and much more compelling dataset obtained from an FFPE sample with many CNAs (see **new Fig. 1k-l and Supplementary Fig. 6 and 7**).

2. In line with the above, the authors show a change in performance of the copy number profile estimation with the diverse settings, such as Figure 1e-g, and Figure 2a-b/d. In all these figures, the authors picked a reference to compare with (e.g., gDNA 30ng with 10 PCR cycles in Figure 1e). However, to prove a gradual decrease in performance due to change of setting (e.g. increased PCR cycle), the authors can provide a correlation heatmap, where row and columns are the setting tested (e.g. amount of input gDNA; similar to Supplementary Figure 4b).

As mentioned above, to convincingly show the performance of the copy number estimation with diverse settings, we have now performed a completely new set of experiments, using gDNA extracted from one FFPE tumor sample that carries many CNAs, and comparing inputs from 1 ng down to 120 pg, and three different PCR cycles (12, 14, and 16). We present the results of these experiments in the **new Fig. 1k-l and Supplementary Fig. 6 and 7**, which show both copy number profiles and plots summarizing inter-sample correlations at various resolutions. In addition, we now provide a thorough quantitative analysis, by showing that the fluctuations of the raw data around the segmented profiles are influenced by various parameters, including sequencing depth (genome coverage), binning size (resolution), and the cutting frequency of the restriction enzyme in use (see **new Supplementary Fig. 3c and 7d-f**).

3. In Figures 1c-d, there is quite some decrease in performance due to the decreased amount of gDNA from 200ng to 100 ng (for instance, A549). After the cell line experiment, many of the tissue-based experiment used small amount gDNA of 30ng. Also, there are often almost no copy number alterations observed for the samples with a small amount of gDNA (Figure 1e, 1f, 2a). We cannot rule out the possibility that CUT-seq is not sensitive enough with low gDNA input. The authors should try a smaller amount of gDNA for cell lines (in figure 1c) to show the robustness of CUT-seq with small amount of gDNA.

These data came from some of the very first experiments using a CUTseq protocol which we have meanwhile substantially improved. As mentioned above, we have now performed a new set of experiments using gDNA extracted from cell lines, using a lower gDNA input (30 ng), as suggested by the Reviewer. As shown in the **new Fig. 1b, c and Supplementary Fig. 3a, b**, the correlations between corresponding HindIII and NlaIII are extremely high for all the cell lines examined, at five different resolutions spanning from 1 Mb up to 30 kb. In addition, we have added a new breast cancer cell line (BT474) and in the **new Fig. 1d** we show that CUTseq specifically detects amplification of the clinically relevant *ERBB2/HER2* gene only in HER2-positive cell lines (BT474 and SKBR3), but not in HER2-negative cells (MCF7).

4. The authors claim that CUT-seq can pick up focal amplifications with 10kb resolution, which is questionable (Figure 1h-i). The difference between signal and noise is not

convincing. It looks quite noisy when focal amplification is called with 10kb resolution, according to the profiles in Supplementary Figure 4a. The authors can cross-check with CNV profile previously reported for SKBR3 cell lines to prove if the noise-like profile is actually the true focal amplification.

We thank the Reviewer for this important consideration. To address this issue, we have now performed a different type of analysis, by examining the reproducibility of DNA copy number profiles at various resolutions, from 1 Mb up to 10 kb, in five pairs of replicate libraries prepared from FFPE tumor samples (previously shown in Fig. 2). As we now show in the **new Fig. 1f, h-j and Supplementary Fig. 5a**, genome-wide copy number profiles are highly correlated between corresponding replicates, even at 10 kb resolution. As the resolution increases, the distribution of the length of amplified and deleted genomic segments progressively shifted towards shorter lengths, in a manner that is reproducible between replicates. Zooming-in on individual chromosomes (see **new Fig. 1j and Supplementary Fig. 5a**) clearly shows that, at high resolution, while the overall copy number profile persists, new features emerge that are reproducibly seen in both replicates. This includes both focal amplifications and deletions, as well as more resolved complex patterns of amplifications and deletions that cannot be seen at lower resolutions.

5. The authors compared CUT-seq with exome capture with the standard Agilent exome-seq protocol. The frequencies and genomic distribution of SNVs are comparable between the two technologies (Figure 2f-g), but also between the two samples (TRN15, TRN 18). It is unclear if the technologies are comparable or the measure comparisons (frequency and genomic distribution) are just insensitive to detect the difference. The authors should perform a systematic and higher-resolution comparison to draw a conclusion. (e.g. how many SNVs captured by both technology, and how comparable they are within and between the samples?)

As already discussed above, our primary motivation to perform this experiment was to show that CUTseq libraries, although prepared in a different way compared to traditional NGS libraries, are compatible with widely used exome-capture kits (e.g., the Agilent SureSelect which we used here). Also, it was never our intention to convey the message that CUTseq is superior to standard methods for WES, and apologize if we did so. In the revised manuscript, we now clearly state this, and highlight the fact that CUTseq reproducibly covers only a fraction of the SNVs that would be normally detected by standard WES methods. In the revised Discussion, we mention a possible application of such reduced representation exome sequencing for studying tumor phylogenies at reduced cost compared to WES. We also discuss the possibility that, by combining two or more restriction enzymes together, the exome coverage of CUTseq could approach the one of standard WES. However, since exome sequencing is not the primary focus of this manuscript, as already mentioned above, if the Reviewer thinks that the clarity of the manuscript would improve if we removed these data, we would of course follow the suggestion and focus exclusively on copy number profiling.

6. The authors explained why they picked the two enzymes, but it is not clear when to use which enzyme. It seems NlaIII is better in terms of coverage and narrower gaps. Is there any benefit of using the other enzyme? Should be discussed in the discussion.

As we now quantitatively show in the **new Supplementary Fig. 3c**, and as it can be seen by comparing HindIII and NlaIII plots at different resolutions, the read count profiles tend to display higher fluctuations in the case of HindIII, especially at higher resolutions. This is due to the fact that, as the size of genomic bins used in the analysis decreases, the number of enzyme recognition sites per bin also diminishes, thus increasing the noise. Since the frequency of recognition sites along the genome is higher for NlaIII compared to HindIII (see **Supplementary Fig. 1**), this explains the observed higher fluctuations for

HindIII profiles. However, it is critical to note that, despite the different noise levels, the segmented profiles are highly stable and still remarkably highly correlated even at 10 kb resolution (see **new Fig. 1f**), indicating that the circular binary segmentation algorithm in use (see Methods) is very robust to such fluctuations. As a rule of thumb, we recommend using a 4-base cutter such as NlaIII when high resolution is desired (<50 kb), otherwise a less expensive 6-base cutter such as HindIII. We have now included these considerations in the revised Discussion.

7. Supplementary Figure 4 a-b do not seem to match with each other. The least correlated pairs should appear at the resolution of 10kb according to 4a, but in 4b, lowly correlated pairs appear at two resolutions (10kb and 50kb, with the lowest sequencing depth). And it seems 10kb resolution gives rather unstable copy number profile.

We have now removed this figure and, as discussed above, we have replaced it with more compelling data and figures.

8. Protocol exchange link (<https://doi.org/10.21203/rs.2.1742/v1>) seems to be not active.

We apologize for the inconvenience. We have checked the following link and it is functional for us:

<https://protocolexchange.researchsquare.com/article/d0ef0512-37b2-461b-9687-eeec11f167e1/v1>

We note that, since Protocol Exchange is part of the Nature Publishing Group, the protocol will be automatically linked to this manuscript (and *vice versa*), should it be accepted for publication in *Nature Communications*. Meanwhile, the protocol can be found in the Supplementary Information.

Reviewer #2

The authors original submission described a novel method, CUTseq, that can be used to perform reduced representation sequencing of DNA from tissue sections or sub-region(s). Isolated DNA is cut with one of a set of CUTseq restriction endonucleases and subsequently ligated to barcoded adapters containing a sample-specific barcode (SB), a unique molecular index (UMI), an Illumina adapter and a T7 promoter. T7 RNA polymerase is then used to transcribe and therefore amplify CUT-seq molecules, which are then converted to a sequencing library by reverse-transcription and PCR. The authors developed the CUTseq method using cell line DNA and further validated it using 17 FFPE samples in which a subset were processed either by CUTseq or conventional DNA library construction.

In my original review of the manuscript, I made a number of comments and I am pleased to say that the authors have addressed each point satisfactorily and together with responses to Reviewer 1, the manuscript is greatly improved. For example, there is 1) greater clarity on the unique benefits of CUTseq; 2) detailed validation data indicating the lower limits (DNA input) of CUTseq and where possible, how it compares with a widely used library method; 3) expected throughput and cost as well as a description/validation of streamlined CUTseq implemented using nano dispensing; 4) a detailed CUTseq protocol resource published on the Protocol Exchange for users wishing to adopt the method.

We are very grateful to the Reviewer for appreciating our efforts to improve the manuscript following both Reviewers' comments and suggestions. In this second revision, we have further corroborated our manuscript, by performing a substantial amount of new experiments and analyses, and by striving to present our results in a clearer way, as already discussed in reply to Reviewer's #1 comments. We hope that the Reviewer will appreciate our amendments and find our manuscript now suitable for publication in *Nature Communications*.

Comment 1: Supplementary Cost analysis - The authors have validated CUTseq on I-DOT and based all their cost calculations using this platform. Could the authors clarify that the commercial alternatives are based on recommendations of the manufacturer, please? It's not the authors responsibility to work out or indeed validate various offerings on I-DOT but the reader should be clear that CUTseq costs are benefiting from low volume dispensing whereas the competition is not.

We have conducted all the cost calculations based on the reagent volumes recommended by the manufacturer of the kits which we took in consideration. In order to make the comparison fairer, we now also include in the Cost Analysis a comparison between CUTseq and NEBNext, without using a nanodispensing device such as I-DOT One. The results of this analysis (see amended **Supplementary Note 1 Figure 1** in the Supplementary Information) show that, even without scaling down CUTseq reagent volumes and simply using a manually operated multi-channel pipette, the cumulative cost still grows significantly faster for all commercially available kits examined compared to CUTseq. We now emphasize in the discussion that the advantage of CUTseq is not bound to the availability of I-DOT, as the method can be readily implemented using standard laboratory equipment.

Comment 2: In this revised manuscript, the authors also demonstrate that CUTseq libraries are compatible with exome sequencing, which will be of particular interest to some groups. However, I am unclear whether the authors now include these data to indicate a compatibility (proof-of-principle) of CUTseq with existing exome capture workflows (Page 8, line 25 - Page 9, line 4 and Page 13, lines 13-15) or to promote CUTseq as a genuinely viable alternative to current exome workflows as suggested on P.13, line 11-12 in particular. I appreciate the

latter point is about choice of enzyme to improve target coverage but I don't feel that there is enough CUTseq performance data presented to assure the reader that the CUTseq combined with exome Seq could be 'virtually indistinguishable from classical exome sequencing methods', even with the perfect restriction enzyme in hand. I just think this point needs clarifying and in its current form (p. 13, para 1) it's potentially misleading.

We thank the Reviewer for raising this important issue. As we also discussed in response to a similar comment by Reviewer #1, it was never our intention to claim that CUTseq is equal or even superior to conventional WES methods, at least when a single restriction enzyme is used. What we aimed to achieve with this experiment, was purely to show that CUTseq libraries are compatible with commercially available kits for exome capture, such as the SureSelect kit from Agilent, which we used here. However, we do acknowledge the fact that, in the first revision, we did not sufficiently emphasize that CUTseq only provides a reduced exome representation (being, by definition, a reduced genome representation method), and did not provide enough evidence for that. We have now performed new experiments, and compared SNVs detected by CUTseq vs. WES in two replicate experiments using the breast cancer SKBR3 cell line. As we show in the **new Fig. 2c-f and Supplementary Fig. 10c-e**, even though the number of high-confidence SNVs identified by CUTseq was lower at comparable sequencing depths— as expected from the fact that CUTseq is a reduced genome representation method — ~38% of the SNVs were found in both CUTseq and WES, and ~72% of all the high-confidence SNVs found by CUTseq were identified in both replicate experiments, further confirming the reproducibility of CUTseq. In the revised Discussion, we now mention that one application of such reduced representation exome sequencing would be in multi-region tumor sequencing, in order to detect less high-confidence SNV events, but from many more regions than currently possible, at comparable sequencing costs. This would significantly improve the ability to reconstruct a tumor's phylogeny, by comparing CNA and SNV profiles from many regions in the same tumor. Furthermore, it is possible that, by combining two or more restriction enzymes, the coverage afforded by CUTseq would increase, approaching the one of standard WES. We would still like to include these results in our final manuscript, since we think they show the full spectrum of CUTseq versatility and applicability. However, as we also wrote to Reviewer #1, if the Reviewer would prefer us to remove this part from the manuscript to improve its clarity, we would of course follow the suggestion.

Comment 3: The authors show that ~80% of the exome target is covered for 2 DNA samples, TRN17 and TRN18. In contrast, the commercial alternative from Agilent targeted ~50%. The results from the Agilent reagents seem disappointing. Notwithstanding the effect of FFPE, I would have expected better from a 50 ng input. Could the authors provide some comment and/or possible explanation? Were the number of reads equivalent, were there enough reads, how do duplicate rates compare?

As discussed above, we have now performed more experiments and also added more quantitative analyses to assess the performance of CUTseq compared to standard methods for WES. The results of these new experiments and analyses are shown in the **new Fig. 2c-f and Supplementary Fig. 10c-e**.

Comment 4: Related to Comment 3 - could the authors indicate in the text the fold (depth) coverage across the target? This will be of interest to users wishing to reliably call variants. Although CUTseq is demonstrated to be effective at ~60 pg input for CNA detection, could the authors estimate/predict lowest input for exome Seq?

We have added information about the coverage in the **new Supplementary Fig. 10d-e**. Regarding the minimum DNA input that could be used for reduced exome representation sequencing by CUTseq, we expect that this will mainly depend on the ability of achieving

a sufficient library yield, to meet the recommendations of commercially available capture kits (e.g., 500 ng of a single library for capture with the SureSelect Agilent kit). While for a single sample of less than a nanogram input DNA it might not be possible to achieve such yield, pooling together multiple samples into the same CUTseq library should be sufficient to achieve the recommended yield. Afterwards, only sequencing depth (i.e., the available budget) will condition how many high-confidence SNVs can be identified. We think that proving this would go beyond the scope of this manuscript, but we are happy to include these considerations in the Discussion, if the Reviewer feels they might be useful to the readership of *Nature Communications*.

Please note that Comments 2-4 and their importance/interest will depend upon what message the authors are wishing to convey with respect to CUTseq/exome Seq. I am convinced that CUTseq is compatible with exome capture based on e.g. Fig 2f-h. but I think anyone genuinely interested in implementing CUTseq with exome capture with the expectation of matching existing methods would benefit from additional performance data as indicated in the above comments.

As said, in this manuscript we indeed only aimed to show a proof-of-principle that CUTseq is compatible with exome capture, and hope that the new experiments and analyses added are sufficient for this purpose.

Comment 5: Fig 3c, heat map - 10 samples were omitted for noisy and therefore unreliable profiles. Could the authors offer an explanation?

We now include all the 88 samples for which there is a sufficient number of usable reads (i.e., aligned reads with the correct prefix after PCR duplicates removal) for reliable copy number calling at 1 Mb resolution (with 100 kb resolution, the number lowers to 86). This is because we deliberately only shallowly sequenced this multiplexed library on a fraction of the NextSeq 500 flowcell, since the primary goal of this experiment was to show the feasibility of high-throughput CUTseq and the reproducibility of copy number profiles across dozens of replicate samples. Thus, as we now show in the **new Figure 3b**, a small number of samples received only few thousands usable reads, even though the vast majority of samples received a homogenous sufficient number of usable reads, typically in the range of 500K–1M. We expect that, if we sequenced the same library deeper, we would be able to obtain reliable copy number profiles also from the missing replicates. However, for obvious economic reasons, we would prefer to avoid this additional experiment, which would anyhow not add significantly to what we already show in Fig. 3.

Comment 6: Supp. Fig. 3d - Duplication rate reaches a plateau (~80%) at ~3ng - any possible explanation?

We have now removed this part, given the fact that all the information about sequencing statistics is provided in the extensive **Supplementary Data 2**, and that many more Supplementary Figures were added to better show copy number profiles, as requested by Reviewer #1. In general, we note that the duplication rate is mainly a function of the number of PCR cycles and gDNA input used, and less of the duration of IVT (which, anyhow, is kept constant to either 14 hours in classical CUTseq or 1.5 hours in high-throughput CUTseq).

Reviewers' Comments:

Reviewer #1:

Remarks to the Author:

The authors significantly improved their manuscript by including much better visualisation and convincing experiments to address all the points raised. Now, the novelty of CUTseq is much more clear, especially in sensitivity to detect copy number alterations and also SNVs. I have a few suggestions, but most of them are minor and I am convinced that the manuscript is competitive enough to be published in Nature Communication.

Specific comments:

1. On page 4, the authors mentioned that: "We selected NlaIII and HindIII based on the frequency and distribution of their recognition sites in the human genome" Not clear how this information is used to make the choice (perhaps enzymes with the average distribution?).
2. It is shown that CUTseq can produce copy number profiles regardless of the choice of enzyme. Though, I think it is important to mention which enzyme was used in FFPE samples in Fig. 1e-f.
3. It is easier for readers if the amount of input gDNA required in commercial kits is given explicitly (instead of referring to supplementary Table 3).
4. Also, the estimated amount of gDNA inputs in FFPE experiments should be mentioned.
5. I strongly recommend to include SNV comparison (Fig. 2c-e). The overlap between SNVs from the two technologies (CUTseq vs Agilent) is high. Though a lower amount of SNVs is detected in CUTseq, high concordance between two replicates indicates high confidence and reproducibility. Also, reduced coverage of SNV detection by CUTseq is nicely explained in the further investigations in supplementary Fig. 10/13 and discussion. I find it very nice and fair.
6. Thresholds for calling amplification/deletion are not given. Necessary to add this in the Method section.
7. I suspect mis-labeling in Supplementary Figure 10c. (CUTseq and Agilent is mixed) Otherwise, the authors need to change their statement in the main text.

Reviewer #2:

Remarks to the Author:

This is now the second revision of the original CUT-seq manuscript (3rd review, therefore). The manuscript has improved greatly since the original submission:

- 1 - The validation data (CUT-seq reproducibility and sensitivity) is now more compelling and certainly more clearly presented (Figs. 1 & 2); the ability to detect clinically relevant copy number changes; reproducibility of copy number changes detected, which is shown to be independent of resolution, PCR cycling and DNA input (125-1000 pg); reliability as determined by benchmarking against a widely used commercially available library preparation reagents.
- 2 - Proof of principle data to demonstrate CUT-seq compatibility with exome capture with a clearly stated caveat that the data generated are at reduced representation.
- 3 - CUT-seq can be streamlined to reduce cost and turnaround time (the authors utilize the I-DOT system to enable streamlining).
4. Demonstrated application of CUT-seq to assess intratumour heterogeneity by profiling DNA copy number changes at high spatial resolution.
5. Detailed supplementary information (supporting 1-4) and methods with a link to Nature protocol exchange.

The original manuscript focused on the development of CUT-seq, its benefits compared to standard library preparation methods and its application to combining histology and NGS analysis to provide spatial information. The first revision included a new development to demonstrate streamlined CUT-seq, most likely in response to reviewers comments that the original submission overstated CUT-seq throughput and scalability. The first revision also added a section to demonstrate that CUT-seq is compatible with exome capture and this section is now far clearer in the second revision.

I believe the authors have responded satisfactorily to the criticisms of the first revision and so I don't feel it is necessary nor fair to enact a new series of changes. Having said that, a comment on the structure that could be considered if it resonates with the other reviewer: The impact/novelty of the manuscript lies in the tech development i.e. the method itself and its validity, its benefits versus the alternative and a clearly demonstrated application that has diagnostic potential. The section on compatibility with exome capture is undoubtedly of interest as is the section on iDOT but these arguably get in the way of the main focus and impact of the paper. The lack of a mention for iDOT/throughput and CUT-seq compatibility with exome Seq in the Abstract aligns with this observation.

One possible solution would be to move exomeSeq and iDOT to something like a 'further developments of CUT-seq' section. I don't believe iDOT was used to generate the data for 'Multi-region copy number profiling' so it doesn't have to come before this section and the exome-Seq section is primarily a 'proof-of-principle'. An alternative solution might be to drop the iDOT to supplementary and boost the exome Seq section to actually reconstruct tumour phylogeny from exome-derived SNV profiles and CNA profiling as per 'Discussion'.

Please note that I am not insisting on re-structuring the manuscript or the additional exome seq data - we all have our own ideas on style and structure after all. However, as mentioned above, if any of my comments align with the thoughts of the other reviewer then the authors may wish to consider the above possible solutions.

Minor comment: Supplementary, Table 5, p. 39 - Beckman not Backman throughout table.

Point-by-point response to referees' comments

Reviewer #1

The authors significantly improved their manuscript by including much better visualisation and convincing experiments to address all the points raised. Now, the novelty of CUTseq is much more clear, especially in sensitivity to detect copy number alterations and also SNVs. I have a few suggestions, but most of them are minor and I am convinced that the manuscript is competitive enough to be published in Nature Communication.

We are very grateful to the Reviewer for appreciating our efforts to further improve our manuscript, and for providing additional suggestions, which we have taken into consideration as summarized in the point-by-point response below.

Specific comments:

1. On page 4, the authors mentioned that: “We selected NlaIII and HindIII based on the frequency and distribution of their recognition sites in the human genome” Not clear how this information is used to make the choice (perhaps enzymes with the average distribution?).

We have now re-phrased this sentence as following:

“We digested the samples using either a more frequent 4-base cutter (NlaIII) or a less frequent 6-base cutter (HindIII). We selected the enzymes among a list of commercially available restriction enzymes that leave staggered DNA ends and are methylation-insensitive (**Supplementary Table 1**), choosing the least expensive enzymes with the most homogeneous distribution of recognition sites in the human genome (**Supplementary Fig. 1a-d**).”

2. It is shown that CUTseq can produce copy number profiles regardless of the choice of enzyme. Though, I think it is important to mention which enzyme was used in FFPE samples in Fig. 1e-f.

We have added this information in the figure legend.

3. It is easier for readers if the amount of input gDNA required in commercial kits is given explicitly (instead of referring to supplementary Table 3).

We have added this information in the main text.

4. Also, the estimated amount of gDNA inputs in FFPE experiments should be mentioned.

We have added this information in the main text.

5. I strongly recommend to include SNV comparison (Fig. 2c-e). The overlap between SNVs from the two technologies (CUTseq vs Agilent) is high. Though a lower amount of SNVs is detected in CUTseq, high concordance between two replicates indicates high confidence and reproducibility. Also, reduced coverage of SNV detection by CUTseq is nicely explained in the further investigations in supplementary Fig. 10/13 and discussion. I find it very nice and fair.

We thank the Reviewer for recognizing that this part is now much more convincing and informative compared to the previous submission. We have been pondering whether to leave this part in Fig. 2, as suggested by the Reviewer, or whether to move it after the description of the multi-region sequencing in breast cancer samples, as suggested by Reviewer #2. However, since we are comparing exome capture performed using CUTseq libraries against capture from libraries made with a commercially available kit (Agilent), we feel that this part is closely related to the CUTseq benchmarking described in Fig. 2a-b. Therefore, we would prefer to leave this part in Fig. 2c-e and only move the part describing high-throughput CUTseq to the last figure, as suggested by Reviewer #1.

6. Thresholds for calling amplification/deletion are not given. Necessary to add this in the Method section.

We have added this information in the Methods section:

“The threshold used for calling amplifications and deletions was $\log_2 \frac{2.5}{2}$ and $\log_2 \frac{1.5}{2}$, respectively.”

7. I suspect mis-labeling in Supplementary Figure 10c. (CUTseq and Agilent is mixed) Otherwise, the authors need to change their statement in the main text.

We thank the Reviewer for spotting this error. We have now corrected the curve colors and legend.

Reviewer #2

This is now the second revision of the original CUT-seq manuscript (3rd review, therefore). The manuscript has improved greatly since the original submission:

- 1 - The validation data (CUT-seq reproducibility and sensitivity) is now more compelling and certainly more clearly presented (Figs. 1 & 2); the ability to detect clinically relevant copy number changes; reproducibility of copy number changes detected, which is shown to be independent of resolution, PCR cycling and DNA input (125-1000 pg); reliability as determined by benchmarking against a widely used commercially available library preparation reagents.
- 2 - Proof of principle data to demonstrate CUT-seq compatibility with exome capture with a clearly stated caveat that the data generated are at reduced representation.
- 3 - CUT-seq can be streamlined to reduce cost and turnaround time (the authors utilize the I-DOT system to enable streamlining).
4. Demonstrated application of CUT-seq to assess intratumour heterogeneity by profiling DNA copy number changes at high spatial resolution.
5. Detailed supplementary information (supporting 1-4) and methods with a link to Nature protocol exchange.

The original manuscript focused on the development of CUT-seq, its benefits compared to standard library preparation methods and its application to combining histology and NGS analysis to provide spatial information. The first revision included a new development to demonstrate streamlined CUT-seq, most likely in response to reviewers comments that the original submission overstated CUT-seq throughput and scalability. The first revision also added a section to demonstrate that CUT-seq is compatible with exome capture and this section is now far clearer in the second revision.

I believe the authors have responded satisfactorily to the criticisms of the first revision and so I don't feel it is necessary nor fair to enact a new series of changes. Having said that, a comment on the structure that could be considered if it resonates with the other reviewer: The impact/novelty of the manuscript lies in the tech development i.e. the method itself and its validity, its benefits versus the alternative and a clearly demonstrated application that has diagnostic potential. The section on compatibility with exome capture is undoubtedly of interest as is the section on iDOT but these arguably get in the way of the main focus and impact of the paper. The lack of a mention for iDOT/throughput and CUT-seq compatibility with exome Seq in the Abstract aligns with this observation.

We thank the Reviewer for very nicely summarizing the improvements to our manuscript from its first submission, and for acknowledging that the current manuscript now convincingly shows the novelty of our method. We fully agree with the Reviewer that the section 'High-throughput CUTseq', which describes an application of the I-DOT robot, stands on the way to the application of CUTseq for multi-region tumor sequencing, in which we did not use I-DOT. Accordingly, following the Reviewer's suggestion, we have now moved this part to the end of the Results section, so that multi-region tumor sequencing is described in Fig. 3. Following Reviewer's #1 comments, we would prefer to leave the part on exome sequencing in Fig. 2, since it involves the comparison with a commercially available method (Agilent SureSelect), and thus is closely related to the

benchmarking part described in Fig. 2a-b. We hope that the Reviewer will find this reasonable.

One possible solution would be to move exomeSeq and iDOT to something like a 'further developments of CUT-seq' section. I don't believe iDOT was used to generate the data for 'Multi-region copy number profiling' so it doesn't have to come before this section and the exome-Seq section is primarily a 'proof-of-principle'. An alternative solution might be to drop the iDOT to supplementary and boost the exome Seq section to actually reconstruct tumour phylogeny from exome-derived SNV profiles and CNA profiling as per 'Discussion'.

As discussed above, following the Reviewer's suggestion, we have moved the I-DOT part to the last figure (shifting the multi-region sequencing part to Fig. 3), while we have left the exome capture part in Fig. 2, in line with Reviewer's #1 comments.

Please note that I am not insisting on re-structuring the manuscript or the additional exome seq data - we all have our own ideas on style and structure after all. However, as mentioned above, if any of my comments align with the thoughts of the other reviewer then the authors may wish to consider the above possible solutions.

We greatly appreciate the Reviewer's balance in providing constructive suggestions, while remaining open to other solutions.

Minor comment: Supplementary, Table 5, p. 39 - Beckman not Backman throughout table.

We have corrected this typo.